# Vamorolone targets dual nuclear receptors to treat inflammation and dystrophic cardiomyopathy

Christopher R Heier[1,2] , Qing Yu[2], Alyson A Fiorillo[1,2], Christopher B Tully[2], Asya Tucker[2], Davi A Mazala[2], Kitipong Uaesoontrachoon[3], Sadish Srinivassane[3], Jesse M Damsker[4], Eric P Hoffman[3,4,5], Kanneboyina Nagaraju[3,4,5], Christopher F Spurney[1,2,6]

**Cardiomyopathy is a leading cause of death for Duchenne muscular dystrophy. Here, we find that the mineralocorticoid receptor (MR) and glucocorticoid receptor (GR) can share common ligands but play distinct roles in dystrophic heart and skeletal muscle pathophysiology. Comparisons of their ligand structures indicate that the Δ9,11 modification of the first-in-class drug vamorolone enables it to avoid interaction with a conserved receptor residue (N770/N564), which would otherwise activate transcription factor properties of both receptors. Reporter assays show that vamorolone and eplerenone are MR antagonists, whereas prednisolone is an MR agonist. Macrophages, cardiomyocytes, and CRISPR knockout myoblasts show vamorolone is also a dissociative GR ligand that inhibits inflammation with improved safety over prednisone and GR-specific deflazacort. In mice, hyperaldosteronism activates MR-driven hypertension and kidney phenotypes. We find that genetic dystrophin loss provides a second hit for MR-mediated cardiomyopathy in Duchenne muscular dystrophy model mice, as aldosterone worsens fibrosis, mass and dysfunction phenotypes. Vamorolone successfully prevents MR-activated phenotypes, whereas prednisolone activates negative MR and GR effects. In conclusion, vamorolone targets dual nuclear receptors to treat inflammation and cardiomyopathy with improved safety.**

## Introduction

Duchenne muscular dystrophy (DMD) is a progressive X-linked disease characterized by muscle degeneration, chronic inflammation, loss of ambulation, and heart failure in the later stages. It is caused by deletion or loss-of-function mutations of the *dystrophin* gene (Monaco et al, 1986; Hoffman et al, 1987; Koenig et al, 1987).

Elevated inflammatory NF-κB signaling is present in infants with DMD, with onset of muscle weakness in early childhood and diagnosis typically made around 5–7 yr of age (Chen et al, 2005). As patients grow older, cardiorespiratory disease develops, and cardiomyopathy is becoming a leading cause of morbidity and mortality (Nigro et al, 1990). Prednisone, an agonist of the glucocorticoid receptor (GR; gene *NR3C1*), is prescribed as a potent anti-inflammatory drug and considered standard of care for DMD (Griggs et al, 2013, 2016; Bello et al, 2015a). Chronic treatment with glucocorticoids leads to a broad range of side effects that detract from patient quality of life, including bone fragility, stunted growth, weight gain, behavior issues, cataracts, and adrenal suppression (Bello et al, 2015a). These negative side effects lead to delay in the age of treatment, poor compliance, and variation in clinical practice (Griggs et al, 2013; Bello et al, 2015a). Deflazacort was recently approved by the Food and Drug Administration (FDA) in efforts to provide a safer alternative to prednisone, after years of use in international and non-DMD populations. However, the approval of deflazacort has become controversial because of large price increases (highlighted by Senator Bernie Sanders in Sanders and Cummings [2017]), along with new reports suggesting it may have increased side effects compared with prednisone (Bello et al, 2015a).

Cardiomyopathy in early adulthood is a leading cause of death for DMD, responsible for 20–50% of mortalities in dystrophin-deficient patients (Eagle et al, 2002; Finsterer & Stollberger, 2003). The earlier, preclinical stages of DMD cardiomyopathy are characterized by normal heart systolic function with developing fibrosis (Hor et al, 2013). DMD myocardial fibrosis is progressive, ultimately leading to dilated cardiomyopathy and systolic dysfunction. Improved clinical management through anti-inflammatory glucocorticoid treatment and respiratory support has substantially improved survival and heart function in patients (Eagle et al, 2002; Silversides et al, 2003; Peterson et al, 2018). Recently, a different class of steroids demonstrated potential for treating DMD cardiomyopathy. These are antagonists of the mineralocorticoid receptor (MR; gene

[1]Department of Genomics and Precision Medicine, George Washington University School of Medicine and Health Sciences, Washington, DC, USA   [2]Center for Genetic Medicine Research, Children's National Medical Center, Washington, DC, USA   [3]AGADA Biosciences Incorporated, Halifax, Nova Scotia, Canada   [4]ReveraGen BioPharma, Incorporated, Rockville, MD, USA   [5]School of Pharmacy and Pharmaceutical Sciences, Binghamton University—State University of New York (SUNY), Binghamton, NY, USA   [6]Division of Cardiology, Children's National Heart Institute, Children's National Medical Center, Washington, DC, USA

Correspondence: cheier@childrensnational.org

NR3C2). MR antagonists such as eplerenone and spironolactone have been used for the treatment of various forms of heart failure and hypertension for decades, and eplerenone is now shown to slow the progression of heart disease in DMD (Raman et al, 2015, 2017).

The GR and MR evolved from a shared ancestral receptor and show ligand cross-reactivity. The GR regulates inflammation and metabolism, whereas the MR regulates fluid and salt levels. The physiological GR hormone cortisol binds both receptors and circulates at higher levels than the MR hormone, aldosterone. To ensure MR signaling specificity, key aldosterone target tissues such as the kidney express the glucocorticoid-inactivating enzyme 11β-hydroxysteroid dehydrogenase 2 (HSD11B2) (Edwards et al, 1988). This enzyme is specifically expressed in epithelial tissues and metabolizes excess glucocorticoids (cortisol and prednisolone) into their inactive forms (cortisone and prednisone) (Funder et al, 1988). In non-epithelial tissues, where this steroid-inactivating enzyme is not expressed (HSD11B2-negative), glucocorticoids can be MR ligands (Edwards et al, 1988; Iqbal et al, 2014). A key non-epithelial MR target tissue in DMD is the heart (Young & Rickard, 2015; Cole & Young, 2017). It is now recognized that MR signaling can impact cardiomyopathy by acting locally, as cardiomyocyte and myeloid MRs promote fibrosis, inflammation, and dysfunction (Young & Rickard, 2015). Increases in MR activity can increase heart mass and promote the progression of pathophysiology in response to additional insults such as high salt or infarction (Brilla & Weber, 1992; Peters et al, 2009; Fraccarollo et al, 2011). Therefore, it would be beneficial to limit the MR cross-reactivity of agonists in HSD11B2-negative tissues such as the heart.

Vamorolone was recently developed as a dissociative GR ligand that provides anti-inflammatory efficacy with improved safety (Heier et al, 2013; Fiorillo et al, 2018; Hoffman et al, 2018). Vamorolone also binds to the MR and has a Δ9,11 structure which avoids enzymes that metabolize cortisol (such as HSD11B2). Here, we investigate the mechanisms of action for vamorolone using studies of MR ligand chemistry, MR reporter assays, and CRISPR knockout myoblasts. We also investigate the effects of MR signaling on dystrophin-deficient hearts using *mdx* mouse models. We report the sensitivity of dystrophin-deficient *mdx* hearts to MR activity, the efficacy of vamorolone as an MR antagonist, and the improved safety of vamorolone versus prednisolone. Our data provide new insights into steroid mechanisms of action, elucidate the molecular pathogenesis of dystrophic cardiomyopathy, and identify vamorolone as a first-in-class drug that targets dual receptors to treat both inflammation and heart failure pathways.

# Results

## Comparison of steroid ligand chemistries

We began to investigate the consequences of MR-binding by the Δ9,11 compound vamorolone by performing in silico studies of the relationships between MR ligand structures, activities, and receptor interactions. By comparing structures of 14 physiological and pharmacological ligands, we found that an 11β-hydroxy group was

only present on MR agonists (Fig 1A). Focusing on a pair of ligands with contrasting effects but similar structures, we found that 11β-hydroxy was the only structural distinction between a potent MR antagonist (progesterone) and MR agonist (11β-hydroxyprogesterone) (Fig 1B). We next queried available X-ray and structural data on ligands bound to their receptors to identify relevant moiety–residue interactions. The structural data showed that the 11β-hydroxy group of 11β-hydroxyprogesterone interacts with MR residue N770 (Fig 1C) through hydrogen bonding (Rafestin-Oblin et al, 2002). Because this residue is conserved between the MR and GR, we next queried whether a conserved interaction also existed between the GR and its ligands. Indeed, the 11β-hydroxy group of dexamethasone has been found to interact with this conserved residue on the GR (N564) through hydrogen bonding (Bledsoe et al, 2002; Hammer et al, 2003; Lind et al, 2000). Supporting its importance in modulating activity, disruption of this conserved interaction by MR or GR mutation (N770A or N564A, respectively) has been shown to maintain ligand binding but disrupt the transcription factor activity of that receptor (Hammer et al, 2003; Rafestin-Oblin et al, 2002). Together, this information indicated that 11β-hydroxysteroids can activate or enhance MR transcription factor functions through interaction with N770. Comparison of vamorolone and prednisolone structures (Fig 1D) provided a situation analogous to that of progesterone and 11β-hydroxyprogesterone, where the key structural difference is the 11β-hydroxy group (Hoffman et al, 2018). Based on these comparisons, vamorolone was anticipated to function as an antagonist of the MR, in direct contrast to prednisolone.

## Identification of vamorolone as an MR antagonist

We next compared the activities of vamorolone, glucocorticoids (prednisolone, deflazacort), aldosterone, and MR antagonists (eplerenone, spironolactone) on receptor activity using MR reporter cell lines (Yang et al, 2015). Testing for MR agonist activity, aldosterone showed a dose-dependent increase in MR reporter activity beginning at a concentration of 1 nM (Fig 1E). Prednisolone also caused a strong dose-dependent activation of the MR, beginning at 10 nM. In contrast, both deflazacort and vamorolone showed no MR agonist activity.

Next, we tested each drug's ability to act as an antagonist that competitively inhibits MR activity. MR reporter cells were pretreated with each drug for 1 h, and MR activity was induced by adding aldosterone (Fig 1F). Vamorolone, eplerenone, and spironolactone inhibited MR activity in a dose-dependent manner. The amount of MR inhibition shown by vamorolone closely resembled that of eplerenone in this in vitro system. Deflazacort showed no inhibition of MR activity. In addition to deflazacort, we also tested the active metabolite 21-desacetyl-deflazacort and found no agonist or antagonist activity (Fig S1). Together, these data indicate that clinical GR ligands differ in their effects on MR activity, and that the GR ligand vamorolone acts as an MR antagonist with potency similar to proven pharmacological antagonists.

## Knockout shows anti-inflammatory activities of ligands are mediated via GR

MR and GR ligands were tested for the ability to induce GR transactivation side effects, and to show efficacy at inhibiting

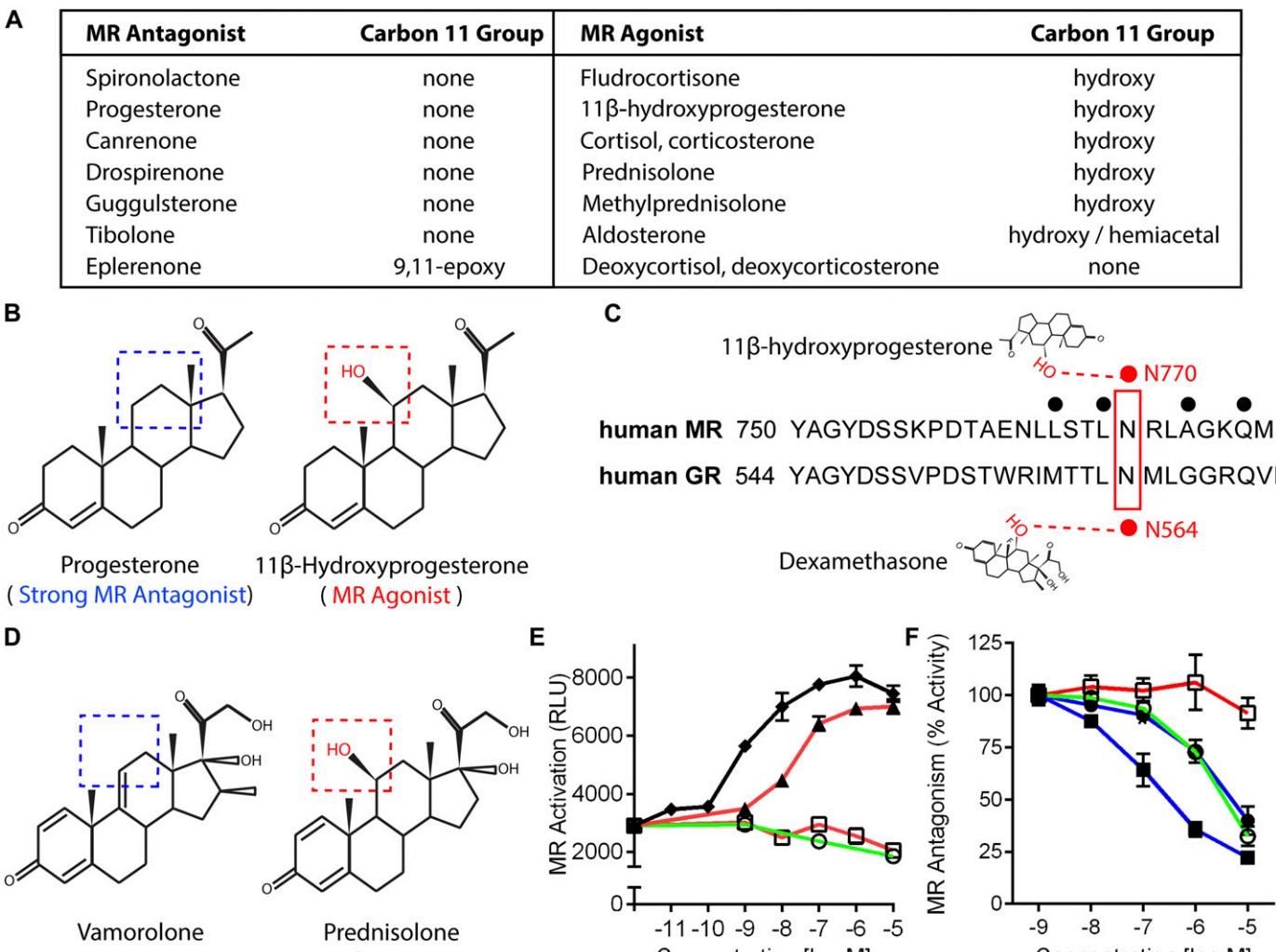

**Figure 1. Vamorolone and MR antagonists lack 11-β hydroxyl groups linked to MR activation.**
**(A)** Table of pharmacological and physiological MR ligands with their carbon 11 group identity provided. **(B)** Progesterone is a potent MR antagonist, whereas addition of an 11β-hydroxy (11β-Hydroxyprogesterone) results in an agonist compound. **(C)** The 11β-hydroxy group of hydroxyprogesterone interacts with MR residue N770 via hydrogen bonding. Dexamethasone also interacts with this conserved residue in the GR (N564) via hydrogen bonding. **(D)** Vamorolone is a Δ9,11 steroid where the 11β position features a carbon–carbon double bond, whereas prednisolone is an 11β-hydroxysteroid. **(E)** A stable MR reporter cell line was treated with drugs and quantified via chemiluminescence assay to determine their agonist properties. Prednisolone and aldosterone showed MR agonist activity. **(F)** Reporter cells were treated with drug in combination with a constant E80 dose of aldosterone to determine antagonist properties. Vamorolone acted as an MR antagonist, consistent with eplerenone and spironolactone. (Representative data from three experiments with each dose performed in triplicate; values are mean ± SEM.)

inflammatory signaling in myogenic cell lines. CRISPR/Cas9 was used to create a GR knockout cell line in C2C12 myoblasts via an N-terminal exon deletion leading to a frameshift (loss of function). After transfection and single-cell clonal propagation, positive clones were identified by PCR and deletions defined by sequencing (Fig 2A–B). Successful elimination of GR protein expression was confirmed by Western blot (Fig 2C).

GR-positive and knockout cells were treated with drugs at equimolar concentrations to determine their transactivation effects on genes directly regulated by a GR-bound promoter element (GRE). This property is associated with negative side effects of current glucocorticoids. For this, we assayed glucocorticoid-induced leucine zipper

(*Gilz*; gene *Tsc22d3*). In wild-type cells, both prednisolone (*P* < 0.05) and deflazacort (*P* < 0.001) caused significant activation of *Gilz* expression (Fig 2D). Vamorolone and eplerenone both showed no significant effects on GR target gene expression. In contrast, and confirming loss of receptor function, GR knockout cells showed no activation of GRE-controlled *Gilz* gene expression by any of the drugs. These data are consistent with the activation of GRE-transactivation side effects by prednisone and deflazacort, but not by vamorolone.

To test for efficacy at inhibiting inflammatory NF-κB signaling, differentiated myotubes were pretreated with drug and then induced with TNF. Inflammatory gene expression was assayed by quantitative reverse transcriptase polymerase chain reaction (qRT-PCR) of

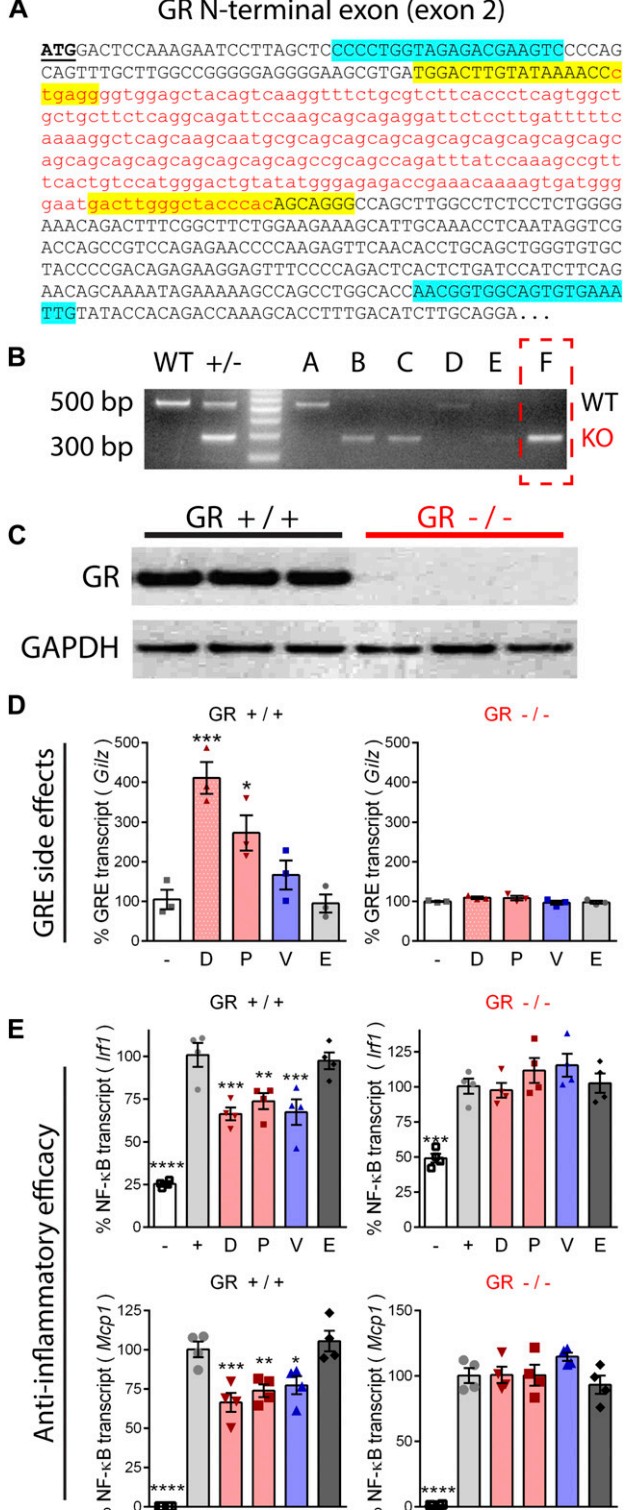

**Figure 2. CRISPR knockout of the GR shows that anti-inflammatory pathways are GR-dependent.**
A frame-shifting deletion mutation was introduced via CRISPR in C2C12 myoblasts to knockout GR expression. **(A)** The DNA sequence of the GR N-terminus is provided. Base pairs deleted in knockout cells are shown as red, lower case letters. Positions of CRISPR guide RNAs are highlighted in yellow, and genotyping primers in blue. **(B)** PCR of C2C12 myoblast clones (labeled A through F) identified

*Interferon regulatory factor 1* (*Irf1*) and *Monocyte chemoattractant protein 1* (*Mcp1*; gene *Ccl2*). These encode an inflammatory transcription factor and a cytokine, respectively, expressed from gene promoters that are directly regulated by NF-κB (Ueda et al, 1994; Robinson et al, 2006). TNF caused a significant increase ($P < 0.0005$) of *Irf1* and *Mcp1* (Fig 2E). Vamorolone, prednisolone, and deflazacort all showed significant anti-inflammatory effects by reducing *Irf1* ($P < 0.005$) and *Mcp1* ($P < 0.05$) in GR-positive cells (Fig 2E). In contrast, the MR-specific drug eplerenone did not show anti-inflammatory effects on *Irf1* or *Mcp1*. In GR knockout cells, no anti-inflammatory effects were seen by any of the drugs, confirming that the GR is specifically responsible for anti-inflammatory effects of vamorolone, prednisolone, and deflazacort. Together, these data indicate that vamorolone alone shows a combined profile of GR-dependent anti-inflammatory effects, selective avoidance of GR side effect pathways, and behavior as an MR antagonist.

### Vamorolone has anti-inflammatory effects in muscle, immune, and heart cells

Our previous experiment demonstrated that GR ligands have anti-inflammatory effects in differentiated myotubes relevant to skeletal muscle pathophysiology in DMD. We next tested drugs for anti-inflammatory efficacy in immune and heart cells, both of which can impact DMD cardiomyopathy. For the first set of experiments, RAW 264.7 macrophages were induced with lipopolysaccharide (LPS) after pretreatment with a GR and/or MR ligand. Analysis by qRT-PCR showed LPS caused a significant increase ($P < 0.0001$) in *Irf1* and *Mcp1*, consistent with data in myotubes (Fig S2). LPS also caused a significant increase ($P < 0.0001$) in *Interleukin 1β* (*Il1b*) and *Interleukin 6* (*Il6*) (Fig 3A). These two cytokines are directly regulated by NF-κB and their chronic overexpression contributes to heart pathophysiology (Libermann & Baltimore, 1990; Hiscott et al, 1993; Wollert & Drexler, 2001; Bujak & Frangogiannis, 2009). Vamorolone, deflazacort, and prednisolone all showed a significant inhibition ($P < 0.005$) of *Irf1*, *Mcp1*, *Il1b*, and *Il6* induction (Figs 3A and S2). The MR-specific drug eplerenone, in contrast, showed no effects on the expression of any of these inflammatory genes. To see if these transcriptional effects were consistent with the levels of cytokines secreted by macrophages, we assayed IL1B and IL6 protein levels in media from the same experiment using an AlphaLISA assay (Fig 3B). The results were consistent with qRT-PCR, showing a potent induction of both secreted IL1B and IL6 with LPS that was significantly attenuated by vamorolone, prednisolone, and deflazacort ($P < 0.0001$ for each), but not by eplerenone.

Next, we tested the ability of GR ligands to inhibit inflammatory signaling in heart cells. We first performed an experiment using

WT (~500 bp band) and GR deletion clones (~300 bp band). **(C)** Western blot confirmed GR knockout in clone from PCR lane F. **(D)** Drug-treated wild-type and GR knockout myoblasts were assayed for activation of a GR transactivation target transcript (*Gilz*). **(E)** Wild-type and GR knockout myotubes were treated with drug, induced with TNF, and assayed for expression of inflammatory genes regulated by NF-κB (*Irf1* and *Mcp1*). ($n = 3$–4, *$P < 0.05$, **$P < 0.005$, ***$P < 0.001$, ****$P < 0.0001$, ANOVA with post hoc versus vehicle [panel D] or + TNF [panel E] control; representative of three experiments; panels [D, E]: [–] = no drug or TNF, [+] = TNF plus vehicle, D = deflazacort, P = prednisolone, V = vamorolone, E = eplerenone).

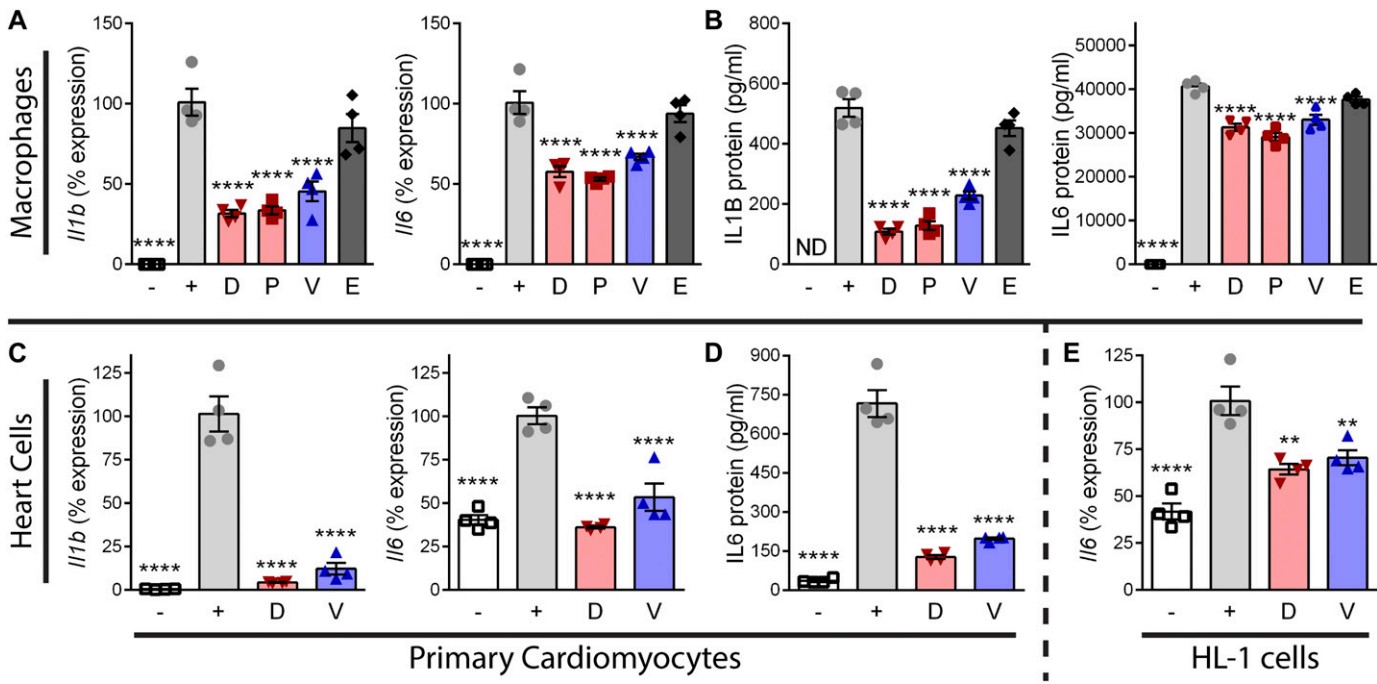

**Figure 3. Vamorolone inhibits inflammatory signaling in macrophage and heart cells.**
**(A)** RAW 264.7 macrophages were pretreated with drug at 10 μM and inflammatory signaling was induced for 24 h using LPS. Expression of NF-κB–regulated inflammatory genes (*Il1b* and *Il6*) was assayed by qRT-PCR. **(B)** IL1B and IL6 protein levels were assayed in media from the same experiment via AlphaLISA assay. **(C)** Primary cardiomyocytes were pretreated with vehicle, vamorolone, or the GR-specific ligand deflazacort, and inflammatory signaling induced with TNF. NF-κB–regulated inflammatory gene expression (*Il1b* and *Il6*) was assayed by qRT-PCR. **(D)** IL6 protein levels were assayed by AlphaLISA. **(E)** HL-1cells were pretreated with 10 μM drug and induced with TNF for 24 h. Expression of *Il6* was assayed by qRT-PCR. (*n* = 4, **$P$ < 0.005, ****$P$ < 0.0001, ANOVA with post hoc versus [+] TNF control in gray; [−] = no TNF control, [+] = TNF plus vehicle, D = deflazacort, P = prednisolone, V = vamorolone, E = eplerenone).

primary cardiomyocytes obtained from postnatal day 2 wild-type mice and treated with either vamorolone or the GR-specific ligand deflazacort. Primary cardiomyocytes displayed spontaneous contractions in culture, characteristic of a heart phenotype, before and throughout treatment. TNF induction caused a significant increase ($P$ < 0.0001) in *Il1b* and *Il6* gene expression (Fig 3C). Administration of vamorolone and deflazacort significantly dampened induction of these genes ($P$ < 0.005). Analysis of IL6 protein levels by AlphaLISA showed consistent results, as TNF caused an increase in IL6 and this was effectively inhibited ($P$ < 0.0001) by both vamorolone and deflazacort (Fig 3D). Next, we repeated drug treatments using HL-1 cells. HL-1 cells are an immortalized cardiac muscle cell line that displays phenotypic characteristics consistent with adult atrial cardiomyocytes (Claycomb et al, 1998). Again, in HL-1 cells TNF increased *Il6* expression ($P$ < 0.0001) and this was successfully decreased ($P$ < 0.005) by vamorolone and deflazacort (Fig 3E). Together, our in vitro data indicate that vamorolone, prednisolone, and deflazacort all possess a GR-dependent ability to inhibit inflammatory signaling in muscle, immune, and heart cells.

### Vamorolone antagonizes the MR in vivo to protect dystrophic hearts

After investigating the mechanism of vamorolone in vitro, we next tested the ability of vamorolone to act as an MR antagonist in vivo. Randomized and blinded treatment groups of wild-type and *mdx*

mice were implanted with subcutaneous osmotic pumps that secreted either vehicle or aldosterone, the physiological MR agonist, at a calculated dose of 0.25 mg/kg/d (*n* ≥ 8 per group). The *mdx* mice receiving aldosterone via osmotic pump were also treated with vehicle, vamorolone (20 mg/kg/d), eplerenone (100 mg/kg/d), or spironolactone (20 mg/kg/d), using daily oral administration via ingestion of sugar syrup formulations. After 6 wk, heart function was assayed by echocardiography, blood pressure was measured, organ weights were obtained, and terminal endpoint measures were performed.

Kidney size was assayed as an aldosterone-specific effect driven by MR activation in epithelial (HSD11B2 positive) tissues. This also provided a readout of MR activity that was independent of direct effects from *mdx* pathophysiology. Aldosterone caused significant increases in kidney size for both wild-type ($P$ < 0.005) and *mdx* ($P$ < 0.0005) mice (Fig 4A and B). In the *mdx* treatment groups, vamorolone, eplerenone, and spironolactone all showed significant protective effects ($P$ < 0.0005), maintaining normal kidney size. This indicates that vamorolone is an efficient MR antagonist in a tissue that is specifically responsive to aldosterone in vivo, consistent with our in vitro data.

Hypertension and cardiomyopathy were assayed as non-epithelial phenotypes that are both related to cardiovascular morbidity. We found that aldosterone caused significant increases in blood pressure for both wild-type ($P$ < 0.005) and *mdx* ($P$ < 0.0005) mice (Fig 5A). In the *mdx* treatment groups, vamorolone caused a

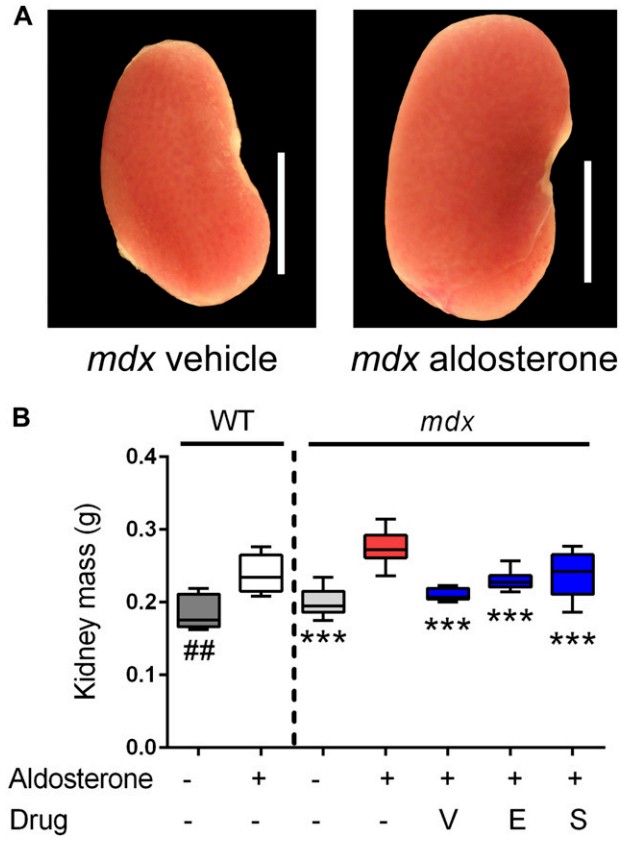

**Figure 4. Vamorolone is an effective MR antagonist in vivo.**
Mice were treated with daily oral drug (vehicle, vamorolone, eplerenone, or spironolactone) and implanted with osmotic pumps that secreted either aldosterone or vehicle for 6 wk. Kidney size was assayed as an epithelial MR target tissue, where aldosterone-specificity of the MR is maintained through expression of the glucocorticoid-inactivating enzyme HSD11B2. **(A)** Representative kidney images are provided. **(B)** Aldosterone induced an increase in kidney mass in both wild-type and *mdx*. This was prevented in *mdx* by all three MR antagonists. ($n ≥ 8$ per group; ***$P < 0.0005$, ANOVA with post hoc test versus *mdx* + aldosterone in red; ##$P < 0.005$, $t$ test comparing WT groups; V = Vamorolone, E = Eplerenone, S = Spironolactone).

significant improvement in blood pressure, as did eplerenone and spironolactone ($P < 0.0005$).

In control groups that received no aldosterone, *mdx* hearts showed expected signs of mild cardiac pathology, including cardiac fibrosis, whereas wild-type hearts did not (Fig 5B and C). Aldosterone exposure in *mdx* mice exacerbated heart fibrosis, causing a significant increase of sirius red staining of collagen deposition in *mdx* hearts ($P < 0.0005$). Vamorolone, eplerenone, and spironolactone all successfully mitigated the development of fibrosis ($P < 0.005$) to no-aldosterone control levels. To examine MR-mediated gene expression changes associated with heart pathology, we used qRT-PCR to assay the expression of MR-regulated genes in heart tissue (Fig 5D). We found that aldosterone significantly increased ($P < 0.05$) expression of *Tenascin C* (*Tnc*), *Fibronectin 1* (*Fn1*), *Matrix metalloproteinase 2* (*Mmp2*), *Collagen 1a1* (*Col1a1*), and *Collagen 6a1* (*Col6a1*). Vamorolone and eplerenone both inhibited the aldosterone-mediated effects on each of these five genes.

Aldosterone exposure additionally caused a large, significant increase in *mdx* heart mass (~30%, $P < 0.005$), whereas in WT mice, it caused a smaller increase that did not reach significance (~10%, $P = 0.15$) (Fig 5E and F). Treatment of *mdx* mice with vamorolone or eplerenone effectively mitigated the increase in heart mass ($P < 0.005$ and $P < 0.05$, respectively). Together, these findings indicate that dystrophin-deficient hearts have increased sensitivity to damage from aldosterone and that vamorolone is effective at preventing this in vivo.

Heart function was assessed using echocardiography after 6 wk of aldosterone exposure (Fig 5G–I). As expected for this age, *mdx* mice did not show a clear decline in heart function as a result of their genotype, as no–aldosterone control wild type and *mdx* were similar in all measures. However, aldosterone caused significant changes in heart function and size which were specific to *mdx* mice. Fractional shortening showed a significant decrease with aldosterone ($P < 0.0005$) exposure (Fig 5G). This decline was successfully improved by treatment with vamorolone, eplerenone, or spironolactone ($P < 0.0005$). Consistent with the increase in heart mass observed in dissections, aldosterone increased left ventricular (LV) wall thickness, in addition to LV mass (Fig 5H and I). Treatment with vamorolone was successful at preventing increases in both wall thickness ($P < 0.05$) and LV mass ($P < 0.005$) toward no-aldosterone controls as well, consistent with eplerenone and spironolactone. In wild-type mice, the only effect observed for aldosterone was a smaller increase in LV mass ($P < 0.05$), which was similar to dissection data (though dissection differences were not significant). Together, these data indicate that MR activation causes a progression of dystrophic cardiomyopathy, whereas vamorolone prevents this progression through a mechanism of MR antagonism.

### Vamorolone improves heart function with improved MR/GR safety versus prednisolone

Prednisolone can act as an MR ligand in non-epithelial tissues (such as heart) that lack expression of the inactivating enzyme HSD11B2. To investigate if this occurs in *mdx* hearts, we first assayed expression of HSD11B2 in heart versus kidney tissue, and then determined if prednisolone activates MR-driven side effects in dystrophic hearts. For the first experiment, we used qRT-PCR to quantify expression of *Hsd11b2* in the kidneys and in the hearts of both wild-type and *mdx* mice (Fig 6A). Consistent with the literature which indicates that HSD11B2 is not expressed in the heart, we did not detect a substantial level of *Hsd11b2* expression in heart tissue from either genotype of mice. Specifically, we found a greater than 10,000-fold decrease in *Hsd11b2* expression in heart tissue in comparison with kidneys ($P < 0.00001$). No significant effect of genotype was observed between *mdx* and wild-type mice for either tissue. These data suggest that kidney MR should be protected from activation by prednisolone, whereas heart MR should be sensitive to activation by prednisolone.

Following this, we obtained archival heart tissue to examine gene expression in the standard *mdx* model of DMD, where no aldosterone exposure was present and mice show a mild phenotype. Previously, we found that prednisolone (5 mg/kg) caused increases in heart mass and fibrosis, whereas vamorolone did not show either of these effects (Heier et al, 2013). Here, we used qRT-PCR to assay

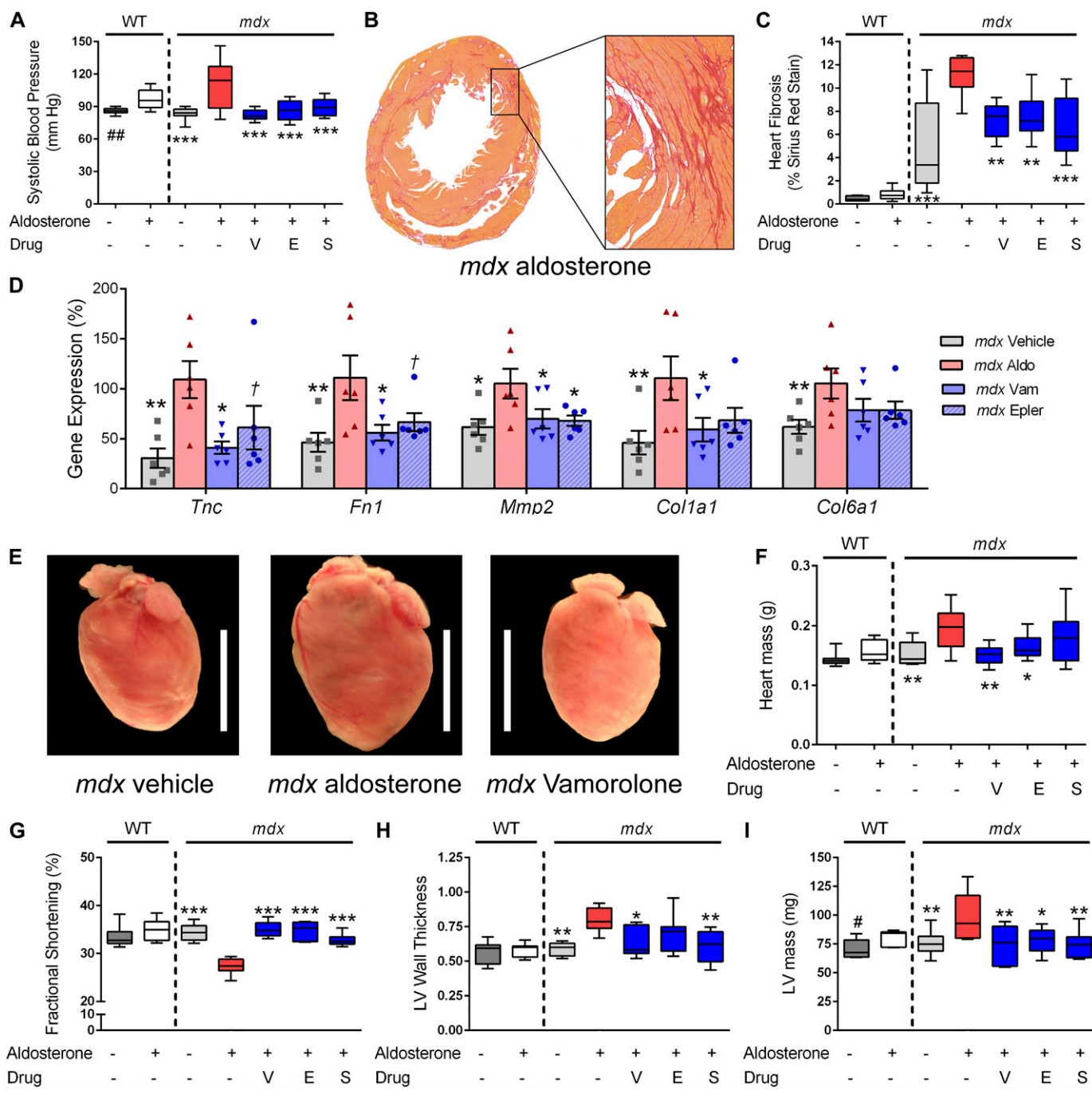

**Figure 5. Vamorolone improves MR-mediated hypertension and *mdx* cardiomyopathy phenotypes.**
Aldosterone introduced via osmotic pumps caused an activation or progression of several phenotypes relevant to cardiomyopathy that were all improved or prevented by vamorolone, eplerenone, and spironolactone. **(A)** Blood pressure was measured by tail cuff, and increased by aldosterone in both wild-type and *mdx* mice. **(B, C)** Sirius Red staining of collagen deposition was used to assay heart fibrosis, where aldosterone increased heart fibrosis specifically in *mdx* mice in (B) representative images and upon (C) quantification. **(D)** qRT-PCR of *mdx* heart tissue was used to assay MR-activated gene expression changes relevant to fibrosis and cardiomyopathy (*n* = 6 per group). **(E, F)** Aldosterone increased *mdx* heart size in (E) representative images and (F) quantitative measurement of heart mass. **(G–I)** Heart function was assayed by echocardiography, with (G) fractional shortening (H) left ventricular wall thickness (LV wall thickness, in millimetre) and (I) left ventricular mass (LV mass) each showing aldosterone-mediated phenotypes that improved with antagonist treatment. (*n* ≥ 8; †*P* < 0.1, *P* < 0.05, **P* < 0.005, ***P* < 0.0005, ANOVA with post hoc versus mdx + aldosterone in red; #*P* < 0.05, ##*P* < 0.005, *t* test comparing WT groups; V = Vamorolone, E = Eplerenone, S = Spironolactone).

the expression of MR-regulated genes in heart tissue (Fig 6B). We found that prednisolone caused significant increases in the expression of *Col6a1*, *Col1a1*, *Tnc*, *Fn1*, and *Mmp2* (*P* < 0.05),

consistent with effects observed for aldosterone in prior experiments. Vamorolone avoided activation of each of these genes.

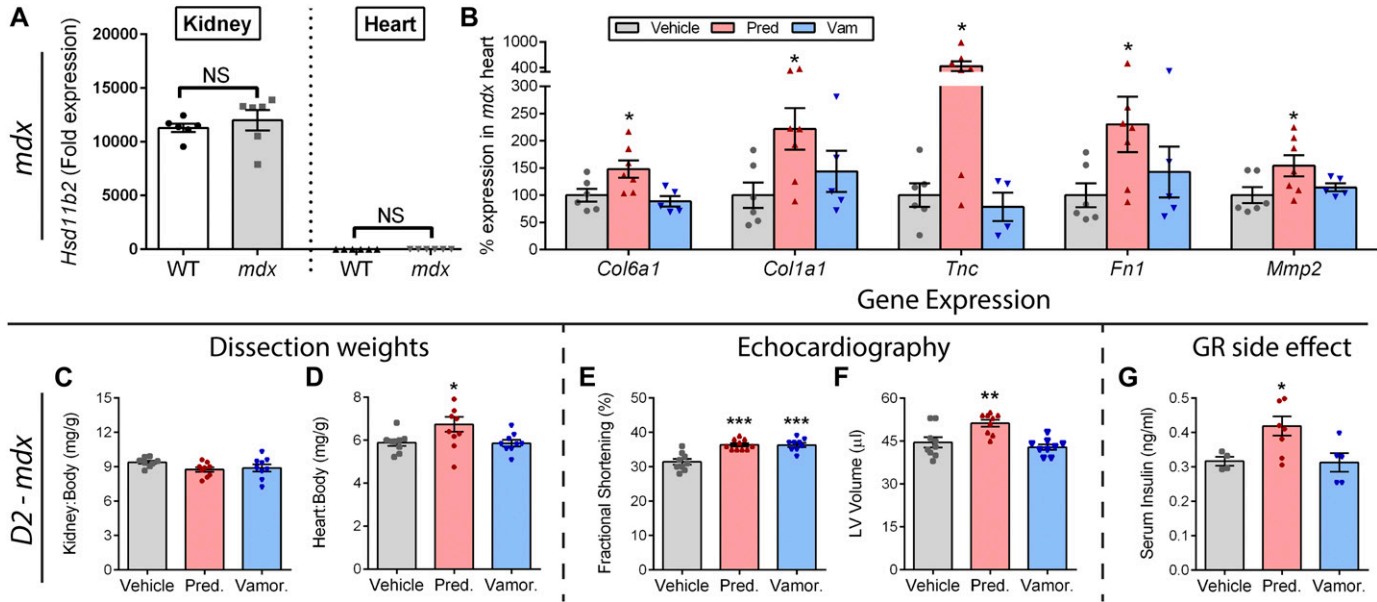

**Figure 6. Prednisolone activates MR and GR side effect pathways in *mdx* and severe *D2-mdx* mice.**
**(A)** Expression of the glucocorticoid-metabolizing gene *Hsd11b2* was assayed by qRT-PCR in kidneys and hearts from wild-type and *mdx* mice. Lack of substantial expression in the heart indicated that the heart is sensitive to MR activation by prednisolone. **(B)** Standard *mdx* mice on a BL10 background were treated with vehicle, prednisolone (5 mg/kg), or vamorolone (45 mg/kg) for 4 months, beginning at 6 weeks of age. The heart was assayed for expression of MR-regulated genes via qRT-PCR and expressed as % vehicle levels ($n \geq 5$). **(C–G)** Severe *D2-mdx* mice on an *LTBP4* modifier background were treated for 8 weeks with daily oral vehicle, vamorolone (30 mg/kg), or prednisolone (5 mg/kg), beginning at 2 mo of age ($n \geq 9$). **(C, D)** Kidney and heart mass at trial conclusion. Note, kidney expresses the glucocorticoid-inactivating enzyme (HSD11B2) that protects it from prednisolone, whereas the heart does not. Prednisolone increased *D2-mdx* heart size but not kidney size. **(E, F)** Echocardiography was performed near the trial endpoint. **(E)** Fractional shortening increased with both vamorolone and prednisolone. **(F)** Left ventricular volume (end diastolic) did not change with vamorolone but increased with prednisolone, consistent with premature dilatation. **(G)** Serum insulin levels increased with prednisolone ($n \geq 5$), consistent with GR-mediated, metabolic side effects of glucocorticoids. ($*P < 0.05$, $**P < 0.005$, $***P < 0.0005$; one panel B outlier removed after significant Grubbs test; (A) t test of *mdx* versus wild-type for each tissue, (B–G) ANOVA with post hoc versus vehicle in gray).

Next, we performed a new trial to determine the effects of daily oral prednisolone versus vamorolone on the heart in a dystrophin-deficient mouse model that exhibits a more severe heart phenotype. Specifically, a genetic modifier shown to be relevant to both human DMD and *mdx* mice is a polymorphism in the *Latent transforming growth factor beta binding protein 4* (*Ltbp4*) gene (Heydemann et al, 2009; Flanigan et al, 2013; Bello et al, 2015b). In mice, the DBA/2J background harbors a deleterious *Ltbp4* genotype. When crossed to the *mdx* dystrophin-deficient strain to produce the *D2-mdx* line of mice, this background significantly worsens *mdx* phenotypes, including an earlier onset of heart dysfunction and cardiomyopathy (Coley et al, 2016). Here, *D2-mdx* mice received daily oral vehicle, prednisolone (5 mg/kg), or vamorolone (30 mg/kg).

Prednisolone treatment showed no effect on kidney size (Fig 6C), but caused a significant increase ($P < 0.05$) in heart size (Fig 6D). This was consistent with expression of the glucocorticoid-inactivating enzyme (HSD11B2) in the kidneys, but not in the heart. No increase in kidney or heart mass was seen with vamorolone treatment. Both vamorolone and prednisolone improved cardiac function ($P < 0.0005$) in *D2-mdx* mice as measured by fractional shortening (Fig 6E). However, prednisolone also caused an increase in LV volume ($P < 0.005$) consistent with an increase in size and early dilatation (Fig 6F). Hyperinsulinemia is a GR-driven side effect induced in human patients treated with prednisone. It is considered a precursor and reliable biomarker for insulin resistance. As expected, prednisolone caused hyperinsulinemia (Fig 6G). Vamorolone did

not induce hyperinsulinemia as a precursor of insulin resistance, which is consistent with recent human Phase 1 data where no changes in blood insulin and glucose levels were observed up through the highest dose (20 mg/kg/d) tested (Hoffman et al, 2018). Together, these data are consistent with activation of MR and GR side effects by prednisolone, whereas vamorolone shows improved safety.

## Discussion

We find that vamorolone targets dual nuclear receptors in a manner that simultaneously treats cardiac disease and chronic inflammation pathways. Vamorolone is a first-in-class MR antagonist/dissociative GR ligand, with MR antagonist potency similar to eplerenone. This has the potential to benefit dystrophic cardiomyopathy, as we find that dystrophin-deficient hearts are specifically sensitive to damage mediated by activated MR. Simultaneously, vamorolone acts as a dissociative GR ligand to provide efficacy as a potent anti-inflammatory with improved safety over the current drugs prednisone and deflazacort. In contrast, eplerenone (MR antagonist) and deflazacort (GR agonist) show more receptor-specific effects, whereas prednisolone acts as a dual MR/GR agonist. A summary of ligand/receptor sub-activities is provided in (Table 1). These data inform both current DMD

**Table 1.** Summary of drug sub-activity profiles on three promoter types for GR and MR ligands.

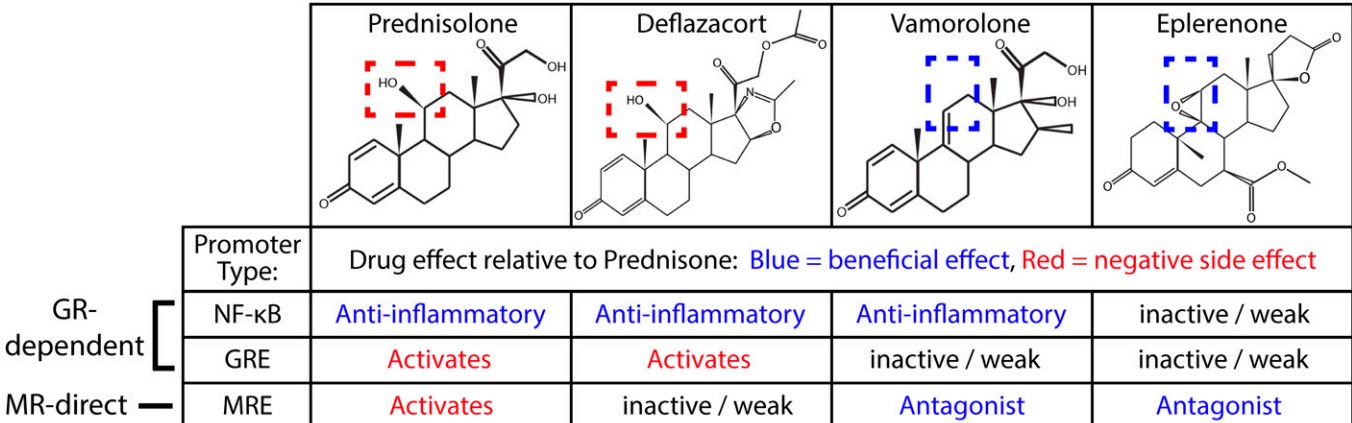

| Promoter Type: | Prednisolone | Deflazacort | Vamorolone | Eplerenone |
|---|---|---|---|---|
| | Drug effect relative to Prednisone: Blue = beneficial effect, Red = negative side effect | | | |
| **GR-dependent** NF-κB | Anti-inflammatory | Anti-inflammatory | Anti-inflammatory | inactive / weak |
| **GR-dependent** GRE | Activates | Activates | inactive / weak | inactive / weak |
| **MR-direct** MRE | Activates | inactive / weak | Antagonist | Antagonist |

treatments and future drug development for a broader group of conditions impacted by steroid treatments.

Steroids are multi-mechanistic and show particular cross-reactivity for the MR and GR. We find vamorolone and prednisolone have contrasting dual-receptor profiles; this has important consequences for heart pathophysiology and is summarized by the model in (Fig 7). The differing activity profiles of vamorolone and prednisolone are consistent with their contrasting structure (Hoffman et al, 2018) and corresponding ability to interact with a conserved residue that modulates receptor transactivation (N770/N564) (Rafestin-Oblin et al, 2002; Hammer et al, 2003). Loss of this transactivation function is likely responsible for the behavior of vamorolone as an MR antagonist, as it can competitively replace MR agonists (aldosterone and prednisolone) with a transactivation-inert ligand. Despite analogous reduction of the GR transactivation function, vamorolone is able to retain its anti-inflammatory efficacy function because it maintains inhibitory protein–protein interactions between the GR and NF-κB (Ratman et al, 2013). In contrast, the 11β-hydroxysteroid prednisolone activates all three of these functions. These findings are consistent with genomic-scale in vivo profiling of the muscle miRNAome, where prednisolone both activates and inhibits miRNA expression, whereas vamorolone exclusively inhibits miRNA expression (Fiorillo et al, 2018). By avoiding off-target transactivation while optimizing two important efficacy pathways, vamorolone can provide both a basic science tool to dissect steroid signaling and a safer therapeutic for inflammatory and heart conditions.

Vamorolone is effective against both hypertension and cardiomyopathy induced by hyperaldosteronism in vivo. Previous mdx cardiac challenges used dobutamine or isoproterenol, both of which stress the heart by stimulating sympathetic nerves to increase the rate and force of cardiac contraction (Danialou et al, 2001; Yasuda et al, 2005; Townsend et al, 2007; Spurney et al, 2011). This worsens cardiomyocyte injury, cardiac pump failure, and acute heart failure. Aldosterone also increases cardiac workload to stress the heart, by regulating blood volume and blood pressure. In addition, aldosterone activates MR signaling locally within specific tissues to increase myocardial fibrosis. Tissue-specific knockout in

cardiomyocytes, macrophages, and endothelial cells show that local MR signaling within these tissues promotes heart inflammation, fibrosis, dilatation, and/or dysfunction (Rickard et al, 2009, 2014; Fraccarollo et al, 2011; Lother et al, 2011; Bienvenu et al, 2012). In the clinic, spironolactone improves severe heart failure at low doses shown to lack diuretic or blood pressure effects (Investigators, 1996; Pitt et al, 1999). Eplerenone, introduced as a safer and more MR-specific (de Gasparo et al, 1987) heart failure drug in 2003, also improves cardiac outcomes in severe and mild forms of heart failure (Pitt et al, 2003; Rossignol et al, 2011; Zannad et al, 2011). Recent DMD studies show that eplerenone improves or slows decline in LV strain, ejection fraction, and myocardial damage, and that these effects are heightened with earlier treatment (Raman et al, 2015, 2017). These findings are consistent with mouse models, where spironolactone used in combination with lisinopril improves cardiac strain and damage in a severe utrophin double mutant ($utr^{+/-}$; mdx) model (Rafael-Fortney et al, 2011; Lowe et al, 2016). Excitingly, MR antagonists continue to evolve as they are relevant to a broader array of dystrophic, hypertensive, and non-hypertensive forms of heart failure.

Eplerenone and deflazacort show increased specificity over the other compounds for the MR and GR, respectively. Eplerenone is part of a series of spironolactone derivatives developed in the 1980's as MR antagonists, with the goal of eliminating spironolactone's off-target side effects on the androgen and progesterone receptors (Smith, 1962; de Gasparo et al, 1989). Modification of the 9,11α-position of spironolactone's structure maintains MR-binding affinity but can improve receptor specificity (de Gasparo et al, 1987). Deflazacort, on the other hand, is an 11β-hydroxysteroid developed as a more targeted anti-inflammatory GR agonist with differentiated tissue-specific bioavailability (Hahn et al, 1981; Diederich et al, 2002). Early reports suggest that it has reduced side effects on bone, adipose tissue, and weight (Mesa et al, 1991; Bonifati et al, 2000; Ferraris et al, 2000; Grosso et al, 2008). We and others find that deflazacort shows increased specificity for the GR, with reduced or absent effects on the MR (Grossmann et al, 2004). This specificity suggests that deflazacort may provide improved safety with regard to MR-specific side effects versus

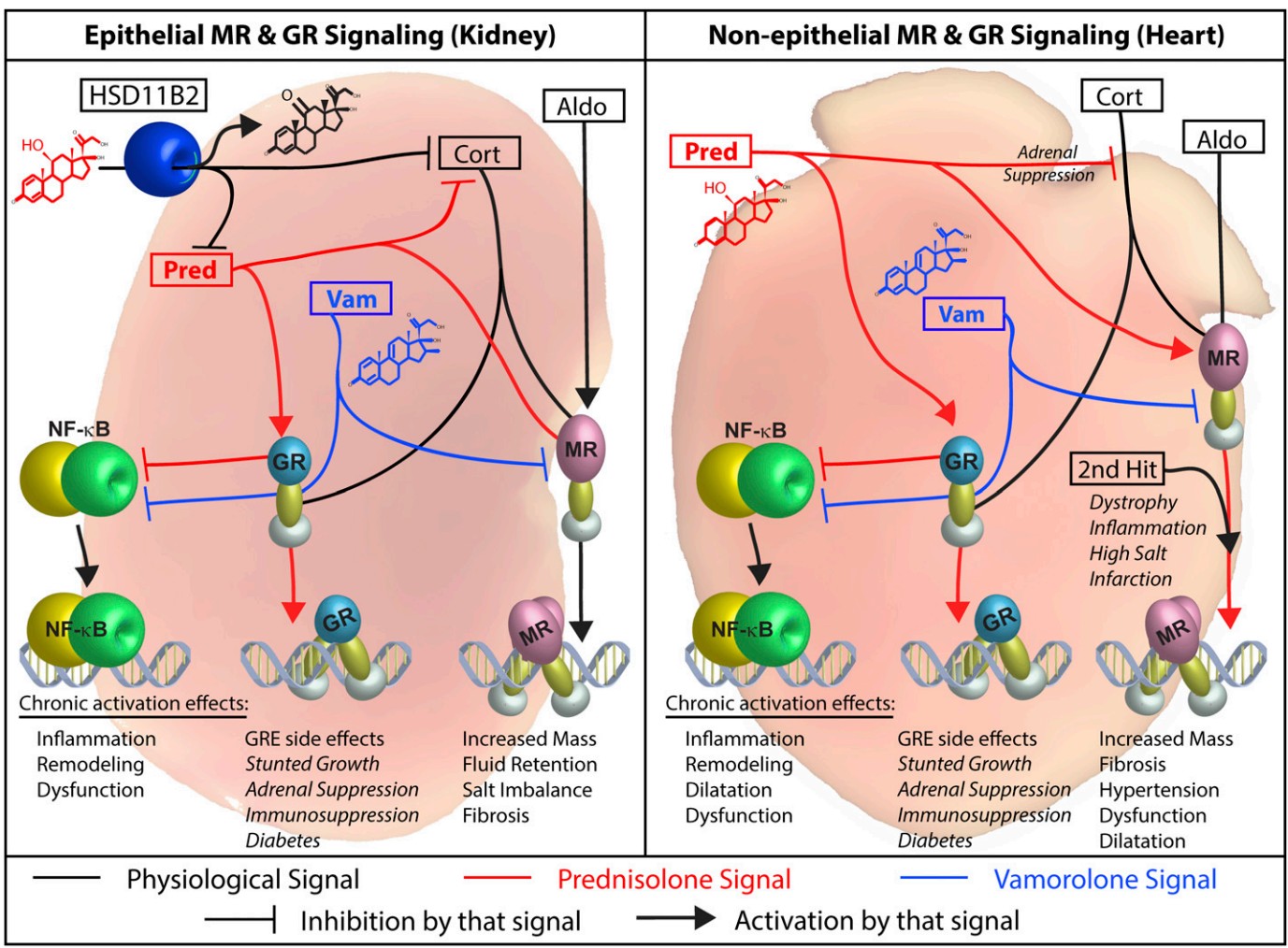

**Figure 7. Working model of dual-receptor drug mechanisms in epithelial and non-epithelial tissues.**
*Left Panel:* Epithelial MR target tissues (e.g., kidney, colon, skin) co-express the HSD11B2 enzyme to prevent over-activation of the MR by glucocorticoids. This protein metabolizes 11β-hydroxysteroids (cortisol and prednisolone) into their inactive ketone states (cortisone and prednisone) within that tissue. Aldosterone is not metabolized by HSD11B2, thus enabling aldosterone-specific signaling in vivo. We find that aldosterone challenge increases kidney size, whereas prednisolone treatment does not, consistent with this model. *Right Panel*: Non-epithelial MR target tissues express the MR but do not express the HSD11B2 enzyme (heart, brain, skin, fat, immune cells). These tissues may have increased risk of damage from MR activation by glucocorticoids. The heart is particularly important here, as (1) DMD patients naturally develop cardiomyopathy, and (2) a "second hit" provided by the DMD disease could worsen MR-mediated heart pathologies (such a "second hit" is seen in other aldosterone models). We find that both aldosterone and prednisolone increase *mdx* heart size and fibrosis, consistent with increased MR activation. *Both Panels*: Vamorolone is a potent MR antagonist that protects both epithelial and non-epithelial tissue types. (Black = physiological ligands, Blue = Vamorolone, Red = Prednisolone; Aldo = Aldosterone, Cort = Cortisol, HSD11B2 = 11β-hydroxysteroid dehydrogenase, Pred = Prednisolone, Vam = Vamorolone).

prednisone, which is a dual GR/MR agonist. However, deflazacort still shows potent GR transactivation, and recent DMD trials report increased cataracts, cushingoid features, and growth delay with deflazacort versus prednisolone (Bello et al, 2015a). Further study should be carried out to dissect MR- from GR-specific side effects, to determine tissue-specific impacts, and to directly compare deflazacort versus prednisone in the DMD clinic.

Future studies using tissue-specific knockout mice and receptor-specific ligands will be valuable to further dissect the roles of the mineralocorticoids, glucocorticoids, their receptors, and their regulatory enzymes. For example, we find that intracellular, inflammatory NF-κB signaling is inhibited by vamorolone, prednisolone, and deflazacort in a GR-specific manner. However,

the renin-angiotensin-aldosterone system also impacts inflammation via effects on reactive oxygen species, blood pressure, blood volume, and inflammatory cell infiltration/adhesion. Accordingly, work by other groups indicate that MR antagonists can also show a level of anti-inflammatory and membrane-stabilizing effects driven independently from the GR and these may provide separate pathways of efficacy for drugs such as eplerenone (Chadwick et al, 2017). In addition, a recent report suggests that myeloid cells can synthesize aldosterone in a manner that could have local impacts on the dystrophic microenvironment of skeletal muscle (Chadwick et al, 2016). In contrast to skeletal muscle in that study, here we do not detect *Hsd11b2* expression in heart tissue; this finding is consistent with our findings that prednisolone can

worsen heart phenotypes of *mdx* mice. It will be intriguing to eventually compare a full dissection of steroidal pathways in muscle, heart, immune, and fibroblast tissue types. Doing so will provide valuable information on how to best apply current clinical drugs, on future drug development, and on the biology of muscle versus heart involvement in DMD.

Prednisolone activation of MR signaling can explain several reports that it has detrimental effects on *mdx* cardiac phenotypes at various stages of disease. At a young age, we find that prednisolone causes premature fibrosis and increased heart mass in *mdx* mice when vehicle controls show no signs of cardiomyopathy (Heier et al, 2013). In older *mdx* mice that naturally present with cardiomyopathy, we find that chronic prednisolone treatment increases heart fibrosis, increases blood pressure, and decreases heart function (Guerron et al, 2010; Heier et al, 2013; Uaesoontrachoon et al, 2014). We should note that there are differences in both heart function and blood pressure between the stages of *mdx* disease; in younger *mdx* here we saw no natural decrease in heart function and no natural blood pressure phenotype, whereas older *mdx* develop a cardiomyopathy with decreased heart function and a decrease in blood pressure (Spurney et al, 2009; Uaesoontrachoon et al, 2014). Other groups also report findings that prednisolone worsens cardiomyopathy in *mdx* mice. A previous article comparing prednisolone to angiotensin-converting enzyme inhibitors finds that administration of prednisolone on its own increases heart fibrosis, LV dilatation, and diastolic dysfunction (Bauer et al, 2009). Recently, two cardiac combination therapies, each containing spironolactone along with another drug, were tested on a background of either no treatment or prednisolone, representing the current standard of care (Janssen et al, 2014). In that study, prednisolone also increases myocardial damage, fibrosis, and strain, whereas this effect was lessened in groups that received spironolactone. Together, these reports are consistent with a model where drugs can impact *mdx* heart phenotypes by affecting MR activity.

We find that dystrophin deficiency is a second hit that specifically sensitizes *mdx* hearts to MR activity. Aldosterone exposure in *mdx* mice causes an increase in heart fibrosis, LV wall thickness, LV mass, and heart mass, with a concurrent decrease in heart function. The corresponding efficacy of MR-specific eplerenone and two other MR antagonists shows that these effects are mediated by activated MR. In contrast, aldosterone exposure by itself shows a lack of progressive heart pathology in wild-type mice, consistent with reports where mice are not additionally exposed to high salt or another insult (Brilla & Weber, 1992; Peters et al, 2009). In the literature, introduction of a "second hit" drives a progression of pathology to cardiac fibrosis and heart failure. This "second hit" is provided in various models by a high salt diet, inflammation, oxidative stress, or myocardial infarction (Brilla & Weber, 1992; Sun et al, 2002; Di Zhang et al, 2008; Fraccarollo et al, 2011; He et al, 2011; Ruhs et al, 2012). We find that the loss of dystrophin in *mdx* mice also provides such a second hit, which could be due to chronic inflammation or increases in cardiomyocyte injury due to the dystrophic phenotype.

Our work here can have both immediate and future clinical impacts. Prednisone, deflazacort, and eplerenone are already being used therapeutically in DMD. Our data strongly support the efficacy of eplerenone in preventing the progression of cardiac disease in DMD, particularly in a background of pharmacological glucocorticoids. Deflazacort shows increased specificity for the GR; however, it maintains GR side effects and lacks the cardiac benefits of MR antagonists. The dual-receptor profile of vamorolone is particularly exciting for its potential to simultaneously treat inflammatory muscle and cardiac pathology with improved safety in DMD. In preclinical animal trials, vamorolone has now been shown to have in vivo anti-inflammatory efficacy as measured by histology, live imaging, cytokine analysis, and/or flow cytometry in models of DMD, inflammatory bowel disease, asthma, limb girdle muscular dystrophy, and multiple sclerosis (Damsker et al, 2013, 2016; Heier et al, 2013; Dillingham et al, 2015; Sreetama et al, 2018). Vamorolone has completed phase I clinical trials in healthy adults, where it showed the loss of most side effects of corticosteroids even through high doses (20 mg/kg/d) (Hoffman et al, 2018). Vamorolone also recently completed phase 2a trials in DMD patients, where it showed anti-inflammatory efficacy and a dissociation of effects on safety versus efficacy biomarkers in patient serum (Conklin et al, 2018). Data from human phase I trials and preclinical animal model trials indicate that vamorolone, in comparison to prednisone, avoids or has substantially reduced side effects on bone, metabolism, adrenal suppression, and immunosuppression. Based on these data, vamorolone has moved forward and is currently being evaluated in phase 2b trials in subjects with DMD. Through shared mechanisms, this has the potential to further impact other diseases characterized by chronic inflammation or cardiovascular pathology.

# Materials and Methods

### MR activity assays

Activity of the MR was assayed via chemiluminescence assay using a stable cell line (PathHunter MR protein interaction assay; DiscoveRx) that expresses two fusion reporter proteins which detect the interaction of activated MR with an MR co-activator protein. To detect MR activation by agonist ligands, cells were treated with increasing concentrations of drug in DMSO for 6 h. To detect inactivation of the MR by antagonist ligands, cells were pretreated with drug for 1 h, then MR activity was induced with aldosterone at an EC80 concentration (2.1 nM) for 5 h.

### CRISPR-mediated GR knockout experiments

CRISPR/Cas9 was used to introduce a deletion mutation into the first coding exon of the GR. A Neon transfection system (Thermo Fisher Scientific) was used to transfect the Cas9 vector into C2C12 myoblasts with gRNA sequences TGGACTTGTATAAAACCCTGAGG and GACTTGGGCTACCCACAGCAGGG. Single cells were isolated on 96-well plates and expanded into homogenous clonal populations. PCR analysis was performed to detect clones with a GR deletion, using forward primer CCCCTGGTAGAGACGAAGTC and reverse primer CAATTTCACACTGCCACCGTT, which produces a 578-bp amplicon for wild-type GR and an approximately 300-bp amplicon for the deletion clones. For subsequent experiments, we selected a knockout

clone with a sequencing-confirmed 271-bp deletion which causes a frameshift and loss of protein expression as detected by Western blot with GR (m-20) antibody (Santa Cruz).

To test the effect of drugs on GR-activated targets, we plated C2C12 and knockout myoblasts in DMEM containing charcoal-stripped FBS overnight, then treated with MR and GR ligands at 100 nM for 24 h. To test the effect of drugs on inflammatory targets regulated by NF-κB, we first differentiated C2C12 cells into myotubes by replacing media with differentiation media (DMEM with 5% horse serum) for 4 d. Myotubes were pretreated with drug for 30 min, followed by the induction of inflammatory transcription by treating with recombinant mouse TNFα (R & D Systems) for 24 h. All cells were lysed in TRIzol for RNA and expression of target genes (*Ccl2, Gilz*, and *Irf1*) quantified by TaqMan Assay (Thermo Fisher Scientific).

### In vitro macrophage and cardiomyocyte experiments

RAW 264.7 cells were cultured in DMEM with 10% FBS. Cells were pretreated with 10 $\mu$M drug. Inflammatory signaling was induced using LPS (Thermo Fisher Scientific) at a dilution of 1:1,000. After 24 h, cells were lysed for RNA using TRIzol and expression of target genes (*Ccl2, Il1b, Il6*, and *Irf1*) quantified by TaqMan Assay (Thermo Fisher Scientific). Levels of secreted proteins were assayed by luminescence-based IL1B and IL6 AlphaLISA assays (PerkinElmer) on media from the same cells assayed by qRT-PCR. Primary cardiomyocytes were isolated from ventricular tissue of hearts in postnatal day 2 C57/BL6 mice by dissecting the ventricles and enzymatically disassociating the cells, with subsequent culture in Cardiomyocyte Maintenance Medium (BrainBits). Cardiomyocytes were observed to display spontaneous contractions in culture, consistent with a cardiac phenotype. Cardiomyocytes were pretreated with 10 $\mu$M drug for 30 min and induced with TNF for 24 h. Expression of *Il1β* and *Il6* was quantified by TaqMan qRT-PCR. IL6 protein was quantified by AlphaLISA (PerkinElmer). HL-1 atrial cardiomyocyte cells (Millipore Sigma) were cultured as previously described (Claycomb et al, 1998) with daily replacement of supplemented Claycomb media (10% FBS, 1× Penicillin/Streptomycin, 0.1 mM norepinephrine, 2 mM L-Glutamine). For treatment, cells were serum starved for 24 h, pretreated with drug for 30 min, and induced with TNF for 24 h.

### Effects of steroid receptor ligands in mice

All animal work was conducted according to relevant institutional, national, and international guidelines, with adherence to standards of the National Institutes of Health (NIH) Guide for the Care and Use of Laboratory Animals. All *mdx* experiments were conducted according to the protocols approved by the Institutional Animal Care and Use Committee of Children's National Medical Center. All *D2-mdx* experiments were conducted according to protocols approved by the University Committee on Laboratory Animals of Dalhousie University. Animals were maintained in a controlled mouse facility with a 12 h light: 12 h dark photoperiod, fed ad libitum, and monitored daily for health. Wild-type (strain 000476; C57BL/10ScSnJ) and *mdx* (strain 001801; C57BL/10ScSn-*Dmd^mdx^*/J) mice were obtained from The Jackson Laboratory. All mice were initially assayed for weight, blood pressure, and echocardiography parameters (*n* ≥ 8

per group). Mice were randomly divided into groups evenly matched for baseline data, with subsequent phenotyping and histology experiments blinded to genotype and drug treatment. For the MR antagonism trial, groups were then pretreated with drug for one week, receiving daily oral drug in cherry syrup (vehicle, vamorolone at 20 mg/kg, eplerenone at 100 mg/kg, or spironolactone at 20 mg/kg). At three months of age, all mice were then implanted with osmotic pumps containing either vehicle (10% ethanol) or aldosterone secreted at a dose of 0.25 mg/kg/d. Drug dosing was continued daily for the duration of the trial. Two mice were removed from the trial, one mouse exhibited aortic stenosis at baseline and was removed at trial onset, the other mouse showed complications from pump implantation surgery and was euthanized. 6 wk after aldosterone pump implantation, mice were assayed for blood pressure and echocardiography. The following week, mice were sacrificed and tissues collected. Histopathology with Sirius Red staining for fibrosis was performed on cross-sections of heart at the level of the papillary muscles. Sirius Red staining was imaged using a VS120 scanning microscope (Olympus) with quantification performed using ImageJ (version 1.48) software.

Expression of *Hsd11b2* was assayed in 5-mo-old *mdx* heart and kidney tissues. Frozen tissues were ground using a liquid nitrogen–cooled mortar and pestle, then RNA was extracted using TRIzol. *Hsd11b2* expression was quantified by TaqMan qRT-PCR assay (Thermo Fisher Scientific).

To examine prednisolone versus vamorolone effects on gene expression linked to MR activation and fibrosis, we assayed archival heart samples. This sample set was taken from a trial in which *mdx* mice were subjected to treadmill running to unmask mild phenotypes, and were treated with either vehicle, prednisolone (5 mg/kg), or vamorolone (45 mg/kg) for four months beginning at 6 wk of age (Heier et al, 2013). In that trial, both drug groups showed improvements in strength measurements and in measurements of inflammation; however, prednisolone caused several side effects that were avoided by vamorolone.

For studies of prednisolone versus vamorolone effects in a more severe model of DMD, the *DBA/2J-mdx* model (or *D2-mdx*) was used. Groups (*n* ≥ 9 per group) of severe *D2-mdx* mice received daily oral vehicle (cherry syrup), vamorolone (30 mg/kg), or prednisolone (5 mg/kg) for a period of 8 wk, starting at 2 mo of age. One mouse from the *D2-mdx* vehicle group was removed from the study and euthanized due to complications from submandibular blood collection during the seventh week of treatment. Echocardiography was performed in the final week of treatment. At treatment endpoint, heart mass was measured and serum collected for measurement of insulin levels by ELISA using an Ultra-Sensitive Mouse Insulin ELISA Kit (Crystal Chem Inc., USA).

### Blood pressure and echocardiography

Mouse blood pressure was measured by tail cuff (Hatteras Instruments). Echocardiography was performed under isoflurane anesthesia using a Vevo 770 micro-ultrasound imaging system in *mdx* and a Vevo 1100 system in *D2-mdx* (VisualSonics). Mice were assayed in M-mode of the parasternal short axis for ejection fraction, fractional shortening, and left ventricular wall thickness at

the level of the papillary muscles. Image analysis and calculations were performed using Vevo software.

## Statistical analysis

The *n* values for each experiment are provided in Figure legends. In vitro studies were analyzed by ANOVA with post hoc multiple comparisons of each treatment with vehicle. Bar graphs represent mean ± SEM. Aldosterone exposure studies were analyzed by ANOVA with post hoc multiple comparisons of each group with aldosterone-exposed *mdx* controls (aldosterone plus vehicle). Gene expression qPCR data were tested for outlier values using a Grubbs test as described in Figure legends. For box and whisker plots, boxes extend from the $25^{th}$ to $75^{th}$ percentile, with the whiskers marking the highest and lowest values for each group. The two wild-type groups (+/− aldosterone) were compared with each other using a two-tailed *t* test. *D2-mdx* studies were analyzed by ANOVA with post hoc multiple comparisons of each treatment to vehicle. Values of *P* < 0.05 were regarded as statistically significant.

# Supplementary Information

# Acknowledgements

The authors would like to thank Marina Moraca, Christina Bell, William Ross, Emma Gillis, and Todd Dow of AGADA Biosciences for their work on treatments and dissections in the *D2-mdx* trial. This work was supported by grants from the Foundation to Eradicate Duchenne, the Clark Charitable Foundation, Action Duchenne, and the NIH. CR Heier and this project were funded by the NIH (K99HL130035, R00HL130035, and L40AR068727). A Fiorillo is supported by the Department of Defense (W81XWH-17-1-047). CR Heier, AA Fiorillo, and DA Mazala were supported by the NIH (T32AR056993). Microscopic analysis for this study was conducted at the Children's Research Institute (CRI) Light Microscopy and Image Analysis Core, which is supported by CRI and the Intellectual and Developmental Disabilities Research Center Award (U54HD090257) through the National Institutes of Health and the National Institute of Child Health and Human Development (NICHD).

## Author Contributions

CR Heier: conceptualization, methodology, investigation, formal analysis, supervision, funding acquisition, and writing—original draft, review and editing.
Q Yu: methodology, investigation, formal analysis, and writing—original draft.
AA Fiorillo: conceptualization, methodology, formal analysis, and writing—original draft, review and editing.
CB Tully: formal analysis, methodology, investigation, and writing—original draft, review and editing.
A Tucker: methodology and investigation.
DA Mazala: Formal analysis, methodology, investigation, and writing—original draft, review and editing.
K Uaesoontrachoon: investigation, formal analysis, and supervision.
S Srinivassane: investigation, formal analysis, and supervision.
JM Damsker: conceptualization and supervision.
EP Hoffman: conceptualization, supervision, funding acquisition, formal analysis, and writing—original draft, review and editing.
K Nagaraju: conceptualization, supervision, formal analysis, and funding acquisition.
CF Spurney: conceptualization, methodology, supervision, funding acquisition, formal analysis, and writing—original draft, review and editing.

## Conflict of Interest Statement

ReveraGen BioPharma owns the method of use intellectual property relating to the use of Δ9,11 compounds to treat disease. EP Hoffman and K Nagaraju are co-founders of ReveraGen with shares in the company. JM Damsker is an employee of the company. The funders had no role in study design, data collection and analysis, decision to publish, or preparation of the manuscript.

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
