## [Reviewer comments · Life Science Alliance]

Life Science Alliance

Vamorolone targets dual nuclear receptors to treat inflammation and dystrophic cardiomyopathy

Christopher Heier, Qing Yu, Alyson Fiorillo, Christopher Tully, Asya Tucker, Davi Mazala, Kitipong Uaesoontrachoon, Sadish Srinivassane, Jesse Damsker, Eric Hoffman, Kanneboyina Nagaraju, and Christopher Spurney

DOI: <https://doi.org/10.26508/lsa.201800186>

Corresponding author(s): Christopher Heier, Children's National Medical Center

Review Timeline:

Submission Date:	2018-08-29
Editorial Decision:	2018-09-25
Revision Received:	2018-12-20
Editorial Decision:	2019-01-18
Revision Received:	2019-01-25
Accepted:	2019-01-28

Scientific Editor: Andrea Leibfried

Transaction Report:

September 25, 2018

Re: Life Science Alliance manuscript #LSA-2018-00186-T

Dr. Christopher R. Heier
Children's National Medical Center
Research Center for Genetic Medicine
111 Michigan Ave NW
Washington, DC 20010

Dear Dr. Heier,

Thank you for submitting your manuscript entitled "Vamorolone targets dual nuclear receptors to treat inflammation and dystrophic cardiomyopathy" to Life Science Alliance. The manuscript was assessed by expert reviewers, whose comments are appended to this letter.

As you will see, the reviewers appreciate your analysis and provide constructive input on how to further strengthen your dataset. We would thus like to invite you to provide a revised version, addressing the concerns of the reviewers. The concerns raised by reviewers #1 and #3 seem straightforward to address. Reviewer #2 asks for more experimental data to substantiate your claims. While we don't expect global gene expression analysis in heart tissues of the mdx mice (rev#2, last point) for acceptance here, we would appreciate an analysis of primary cardiomyocytes/another relevant cell type as proposed by this reviewer. We are happy to discuss this point further with you should this be helpful.

Thank you for this interesting contribution to Life Science Alliance. We are looking forward to receiving your revised manuscript.

Sincerely,

Andrea Leibfried, PhD
Executive Editor
Life Science Alliance
Meyershofstr. 1
69117 Heidelberg, Germany
t +49 6221 8891 502
e a.leibfried@life-science-alliance.org
www.life-science-alliance.org

- A letter addressing the reviewers' comments point by point.
- An editable version of the final text (.DOC or .DOCX) is needed for copyediting (no PDFs).
- High-resolution figure, supplementary figure and video files uploaded as individual files: See our detailed guidelines for preparing your production-ready images, <http://life-science-alliance.org/authorguide>
- Summary blurb (enter in submission system): A short text summarizing in a single sentence the study (max. 200 characters including spaces). This text is used in conjunction with the titles of papers, hence should be informative and complementary to the title and running title. It should describe the context and significance of the findings for a general readership; it should be written in the present tense and refer to the work in the third person. Author names should not be mentioned.

B. MANUSCRIPT ORGANIZATION AND FORMATTING:

Full guidelines are available on our Instructions for Authors page, <http://life-science-alliance.org/authorguide>

Reviewer #1 (Comments to the Authors (Required)):

This manuscript by Heier and Spurney reports a very nice series of analyses that dissect the differential effects of vamorolone versus prednisolone, deflazacort, eplerenone, and spironolactone on activating/inhibiting mineralcorticoid and glucocorticoid receptors mediating the inflammation

and cardiomyopathy associated with dystrophinopathy in the mdx mouse model. The experiments are presented in a clear and balanced manner. I think the study is important to the field and will be widely read. Below, however, are a few issues that should be addressed.

1) The data in Fig. 4A appear to report that systolic blood pressure in mdx mice that received no aldosterone was not different from WT controls. However, previous studies from this group have reported that mdx mice have significantly lower systolic (CF Spurney et al, K Uaesoontrachoon et al., 2014), diastolic (CF Spurney et al, 2009), or mean blood pressure (CF Spurney e. al, 2009) compared to WT. Can they explain why blood pressure was not lower in the mdx controls compared to WT? Perhaps the current experiment is underpowered to reproduce their previous findings?

2) The bar graph data in Fig. 2D, 2E, 4D, and 5A-F could be more easily evaluated by readers if presented as dot plots.

3) The statement in line 4 is not supported by the reference provided.

Reviewer #2 (Comments to the Authors (Required)):

Treatment with glucocorticoids of patients with Duchenne muscular dystrophy (DMD) is considered as standard of care. The chronic use of glucocorticoids has been shown to substantially preserve heart function and improve survival of DMD patients. In the present study, Heier et al. aim to identify the underlying mechanism(s) of Vamorolone, a glucocorticoid receptor (GR) ligand that has been developed from the same group (Heier et al., 2016 EMBO Molecular medicine).

Overall, the concept and results of this manuscript are presented in a very structured manner. The conclusion that vamorolone mediates both anti-inflammatory and anti-fibrotic effects in DMD hearts, however, has to be substantiated by additional methodological approaches. Likewise, the quantification of miR-146 alone is not sufficient to determine anti-inflammatory activities of vamorolone and other drugs being studied here. The use of C2C12 myoblasts is furthermore not the best model to study anti-inflammatory effects of vamorolone, which is supposed to act on local cells of the injured heart. Cultures of primary cardiomyocytes, fibroblasts or even immune cells that are known to invade and accumulate during heart failure progression (e.g. macrophages) should be used instead as an in vitro to study direct effects of vamorolone. A more detailed analysis of myocardial samples assessing the degree of inflammation (flow cytometry, histology) and fibrosis is also recommended.

Minor comments:

- The introduction is too long and contains partly repetitions of content.
- The steroidal drugs and metabolic derivatives enlisted within the abstract (vamorolone, prednisolone, prednisone, deflazacort) should be clearly referred to its corresponding receptor system they activate (if possible).
- The figure legends should not contain a description and interpretation of the results itself, e.g. "Vamorolone acted as an MR antagonist, consistent with eplerenone and spironolactone" (Fig. 1) and "Prednisolone caused an increase in D2-mdx heart size" (Fig.5).
- Could the authors estimate the amount of vamorolone per day that the mice received upon osmotic pump implantation? A clear statement is missing within the results section.
- The antagonistic effect of vamorolone on MR activation in vivo is restricted to maintain the kidney size upon treatment with aldosterone. This aspect of vamorolone is not directly linked to its mode of action at sites of cardiac dystrophy and thus should be transferred to a supplemental material section.
- It is stated within the result section "Vamorolone shows potent MR antagonist activity in vivo"

that "After six weeks, heart function was assayed by echocardiography, blood pressure was measured, and terminal endpoint measures were performed." These data, however, are not presented here.

- The authors should employ a more unbiased and global gene expression profiling approach than selective qRT-PCR to characterize changes of gene expression in heart tissues of mdx and valmorolone treated mdx mice. Such an approach would also reflect potential alterations of the degree of inflammation.

Reviewer #3 (Comments to the Authors (Required)):

This manuscript provides a comprehensive set of well designed and conducted studies demonstrating that valmorolone works as an antagonist of the mineralocorticoid receptor (MR) in heart in addition to an agonist of the glucocorticoid receptor (GR). These findings are important due to the ongoing clinical trials with valmorolone for Duchenne muscular dystrophy as well as the current standard of clinical care, which are GR agonists. In silico analyses, reporter assays, GR knockout myoblasts, analysis of GR and MR target genes, are shown to demonstrate the molecular mechanisms of valmorolone's action. In vivo studies of the effect of valmorolone versus eplerenone, and spironolactone on aldosterone treated mdx mice and valmorolone versus prednisolone treatment of D2-mdx mice were compared were used to further demonstrate valmorolone's action as an MR antagonist. Overall, this is a well- conducted study and a well written manuscript.

Two minor points in the discussion should be considered. The authors should be careful about their assumption that the glucocorticoid inactivating enzyme HSD11B2 is not expressed in dystrophic heart, since others have shown this enzyme appears to be increased in mdx skeletal muscles (Hum Mol Genet. 2016 Dec 1;25(23):5167). Although it is reasonable that this experiment may be beyond the scope of the current study, the authors should consider both possibilities and be wary of drawing a model that compares cell types with and without this enzyme. It is possible that the upregulation of HSD11B2 actually explains why dystrophin deficiency is the "second hit" that sensitizes mdx hearts to MR activity. Second, in vivo anti-inflammatory gene expression changes similar to prednisone have also been observed with other MR antagonists and could be referenced (Am J Physiol Cell Physiol. 2017 Feb 1; 312(2): C155). It is possible that although the specific target miRNA assessed in these experiments is GR-dependent, that other inflammatory genes come from MR activation.

Reviewer points and responses:

Reviewer #1:

This manuscript by Heier and Spurney reports a very nice series of analyses that dissect the differential effects of vamorolone versus prednisolone, deflazacort, eplerenone, and spironolactone... The experiments are presented in a clear and balanced manner. I think the study is important to the field and will be widely read. Below, however, are a few issues that should be addressed.

A) The statement in line 4 is not supported by the reference provided.

- We corrected this by double-checking references and adding the following references:
 - References added to line 4 (page 3) in the Introduction:
 - Koenig et al. 1987, *Cell*
 - Monaco et al. 1986, *Nature*
- There was another reference for a Line 4 on page 9 of the results that may have been referred to, however this was removed when that section was changed in response to a Reviewer # 3 point.

B) The bar graph data in Fig. 2D, 2E, 4D, and 5A-F could be more easily evaluated by readers if presented as dot plots.

- We reformatted these graphs to present dots plotted for the data points in this revised version of our manuscript.

C) The data in Fig. 4A appear to report that systolic blood pressure in mdx mice that received no aldosterone was not different from WT controls. However, previous studies from this group have reported that mdx mice have significantly lower systolic (CF Spurney et al, K Uaesoontrachoon et al., 2014), diastolic (CF Spurney et al, 2009), or mean blood pressure (CF Spurney et al, 2009) compared to WT. Can they explain why blood pressure was not lower in the mdx controls compared to WT?

- We have added text addressing this to our Discussion.
- We believe the reason for the difference is due to the difference in *mdx* mouse ages / diseases stages between the studies. In our current study we use younger mice that are pre-symptomatic in terms of cardiomyopathy. In those other studies, we used older *mdx* mice (≥ 10 months of age) which are at an advanced stage of *mdx* disease where they display symptomatic cardiomyopathy. For the experiments in our current paper, our goal was to introduce aldosterone challenge onto a background of pre-symptomatic disease stage of *mdx* cardiomyopathy. This was intentionally done by design; we wanted to study a disease stage where aldosterone could effectively exacerbate mild / pre-symptomatic disease to a more severe / symptomatic state, while also testing the ability of antagonists to counteract its effects to keep disease in a mild cardiac phenotype.
- Our study had a similar number of mice to the provided references (n = 8-10 per group), so we do not believe differences were due to a difference in power.

- Text added to the Discussion:
 - “We should note there are differences in both heart function and blood pressure between stages of *mdx* disease; in younger *mdx* here we saw no natural decrease in heart function and no natural blood pressure phenotype, while older *mdx* develop a cardiomyopathy with decreased heart function and a decrease in blood pressure (Spurney et al, 2009; Uaesoontrachoon et al, 2014).”

Reviewer #2:

This manuscript provides a comprehensive set of well designed and conducted studies demonstrating that valmorolone works as an antagonist of the mineralocorticoid receptor (MR) in heart in addition to an agonist of the glucocorticoid receptor (GR). These findings are important due to the ongoing clinical trials with valmorolone for Duchenne muscular dystrophy as well as the current standard of clinical care, which are GR agonists... Overall, this is a well-conducted study and a well written manuscript.

Two minor points in the discussion should be considered.

A) The authors should be careful about their assumption that the glucocorticoid inactivating enzyme HSD11B2 is not expressed in dystrophic heart, since others have shown this enzyme appears to be increased in *mdx* skeletal muscles (Hum Mol Genet. 2016 Dec 1;25(23):5167). Although it is reasonable that this experiment may be beyond the scope of the current study ...

- These were interesting studies (references in points A & B) and we have added the two references from this group (Chadwick et al. 2017 and Chadwick et al. 2016) along with text discussing them in the context of our study to our Discussion. Text added to the Discussion is found in our Response to point B.
- Although the Reviewer said “this experiment may be beyond the scope of the current study”, we also decided to assay HSD11B2 expression in both WT and *mdx* hearts.
 - We found and now show HSD11B2 is not substantially expressed in heart tissue from either WT or *mdx* mice. We have added this data to the paper (Figure 6A) along with relevant text to the Results and Discussion sections.

B) In vivo anti-inflammatory gene expression changes similar to prednisone have also been observed with other MR antagonists and could be referenced (Am J Physiol Cell Physiol. 2017 Feb 1; 312(2): C155). It is possible that although the specific target miRNA assessed in these experiments is GR-dependent, that other inflammatory genes come from MR activation.

- We added the Reviewer references, and text discussing them to the Discussion to address Reviewer points A and B:
 - “Future studies using tissue-specific knockout mice and receptor-specific ligands will be valuable to further dissect roles of the mineralocorticoids, glucocorticoids, their receptors and their regulatory enzymes. For example we find intracellular, inflammatory NF- κ B signaling is inhibited by vamorolone, prednisolone and deflazacort in a GR-specific manner. However, the renin-angiotensin-aldosterone system (RAAS) also impacts inflammation via effects on reactive oxygen species, blood pressure, blood volume, and inflammatory cell infiltration/adhesion. Accordingly, work by other groups indicates MR antagonists can also show a level of anti-inflammatory and membrane-stabilizing effects driven independently from the GR and these may provide separate pathways of efficacy for drugs such as eplerenone (Chadwick et al, 2017). Additionally, a recent report suggests myeloid cells can synthesize aldosterone in a manner that could have local impacts on the dystrophic microenvironment of skeletal muscle (Chadwick et al, 2016). In contrast to skeletal muscle in that study, here we do not detect *Hsd11b2* expression in heart tissue; this finding is consistent with our findings that prednisolone can worsen heart phenotypes of *mdx* mice. It will be intriguing to eventually compare a full dissection of steroidal pathways in muscle, heart, immune and fibroblast tissue types. Doing so will provide valuable information on how to best apply current clinical drugs, on future drug development, and on the biology of muscle versus heart involvement in DMD.”

Reviewer #3:

Overall, the concept and results of this manuscript are presented in a very structured manner. The conclusion that vamorolone mediates both anti-inflammatory and anti-fibrotic effects in DMD hearts, however, has to be substantiated by additional methodological approaches.

A) The quantification of miR-146 alone is not sufficient to determine anti-inflammatory activities of vamorolone and other drugs being studied here. The use of C2C12 myoblasts is furthermore not the best model to study anti-inflammatory effects of vamorolone, which is supposed to act on local cells of the injured heart. Cultures of primary cardiomyocytes, fibroblasts or even immune cells that are known to invade and accumulate during heart failure progression (e.g. macrophages) should be used instead as an in vitro to study direct effects of vamorolone.

- We performed and added new experiments assaying anti-inflammatory effects in Primary Cardiomyocytes isolated from Postnatal Day 2 mouse hearts.
 - In parallel, we performed and added a new experiment assaying anti-inflammatory effects in the immortalized HL-1 cell line derived from cardiomyocytes.

- We performed and added new experiments assaying anti-inflammatory effects in macrophages.
- Data from these added experiments is now presented in Figure 3 and the Results.
 - Figure 3:

Figure 3: Vamorolone inhibits inflammatory signaling in macrophage and heart cells. (A) RAW 264.7 macrophages were pretreated with drug at 10 μ M and inflammatory signaling induced for 24 hrs using LPS. Expression of NF- κ B regulated inflammatory genes (*Il1b* and *Il6*) was assayed by qRT-PCR. (B) IL1B and IL6 protein levels were assayed in media from the same experiment via AlphaLISA assay. (C) Primary cardiomyocytes were pretreated with vehicle, vamorolone, or the GR-specific ligand deflazacort, and inflammatory signaling induced with TNF. NF- κ B regulated inflammatory gene expression (*Il1b* and *Il6*) was assayed by qRT-PCR. (D) IL6 protein levels were assayed by AlphaLISA. (E) HL-1 cells were pretreated with 10 μ M drug and induced with TNF for 24 hrs. Expression of *Il6* was assayed by qRT-PCR. (n = 4, ** p < 0.005, **** p < 0.0001, ANOVA with post-hoc vs. (+) TNF control in gray; (+) = TNF plus vehicle, D = deflazacort, P = prednisolone, V = vamorolone, E = eplerenone)

- New text added to Results:

- “We next tested drugs for anti-inflammatory efficacy in immune and heart cells, both of which can impact DMD cardiomyopathy. For the first set of experiments, RAW 264.7 macrophages were induced with lipopolysaccharide (LPS) after pre-treatment with a GR and/or MR ligand. Analysis by qRT-PCR showed LPS caused a significant increase (p < 0.0001) in *Irf1* and *Mcp1*, consistent with data in myotubes (Supplementary Fig S2). LPS also caused a significant increase (p < 0.0001) in *Interleukin 1 β* (*Il1b*) and *Interleukin 6* (*Il6*) (Fig 3A). These two cytokines are directly regulated by NF- κ B and their chronic overexpression contributes to heart pathophysiology (Bujak & Frangogiannis, 2009; Hiscott et al, 1993; Libermann & Baltimore, 1990; Wollert & Drexler, 2001). Vamorolone, deflazacort and prednisolone all showed a significant inhibition (p < 0.005) of *Irf1*, *Mcp1*, *Il1b* and *Il6* induction (Figs 3A and S2). The MR-specific

drug eplerenone, in contrast, showed no effects on the expression of any of these inflammatory genes. To see if these transcriptional effects were consistent with the levels of cytokines secreted by macrophages, we assayed IL1B and IL6 protein levels in media from the same experiment using an AlphaLISA assay (Fig 3B). Results were consistent with qRT-PCR, showing a potent induction of both secreted IL1B and IL6 with LPS that was significantly attenuated by vamorolone, prednisolone, and deflazacort ($p < 0.0001$ for each), but not by eplerenone.

Next, we tested the ability of GR ligands to inhibit inflammatory signaling in heart cells. We first performed an experiment using primary cardiomyocytes obtained from postnatal day 2 wild type mice and treated with either vamorolone or the GR-specific ligand deflazacort. Primary cardiomyocytes displayed spontaneous contractions in culture, characteristic of a heart phenotype, before and throughout treatment. TNF induction caused a significant increase ($p < 0.0001$) in *Ilf1b* and *Ilf6* gene expression (Fig 3C). Administration of vamorolone and deflazacort significantly dampened induction of these genes ($p < 0.005$). Analysis of IL6 protein levels by AlphaLISA showed consistent results, as TNF caused an increase in IL6 and this was effectively inhibited ($p < 0.0001$) by both vamorolone and deflazacort (Fig 3D). Next, we repeated drug treatments using HL-1 cells. HL-1 cells are an immortalized cardiac muscle cell line that displays phenotypic characteristics consistent with adult atrial cardiomyocytes (Claycomb et al, 1998). Again, in HL-1 cells TNF increased *Ilf6* expression ($p < 0.0001$) and this was successfully decreased ($p < 0.005$) by vamorolone and deflazacort (Fig 3E). Together, our *in vitro* data indicate vamorolone, prednisolone and deflazacort all possess a GR-dependent ability to inhibit inflammatory signaling in muscle, immune, and heart cells. “

- Skeletal Muscle inflammation also remains a key issue and drug target for DMD. In considering the Reviewer comments on a microRNA readout in myoblasts, we also replaced our prior experiment (Figure 2E) by performing a new experiment using differentiated myotubes where we assay drug effects on messenger RNA from more established NF-κB target genes.

- New Figure 2E:

- B) A more detailed analysis of myocardial samples assessing the degree of inflammation (flow cytometry, histology) and fibrosis is also recommended.

- In the revised manuscript, histological examination of heart fibrosis is provided in Figure 5B-C.

○

○

- In the revised manuscript, gene expression analyses relevant to fibrosis and cardiomyopathy are provided in Figure 5D.

○

- We have now characterized the *in vivo* anti-inflammatory efficacy of Vamorolone quite extensively in several different animal models, and have shown a successful decrease in serum-based inflammatory biomarkers in DMD patients in Phase 2a clinical trials.

○

In the revised manuscript, we now add text summarizing this data in our Discussion section. Text added to Discussion:

- “In preclinical animal trials, vamorolone has now been shown to have *in vivo* anti-inflammatory efficacy as measured by histology, live imaging, cytokine analysis, and/or flow cytometry in models of DMD, inflammatory bowel disease, asthma, limb girdle muscular dystrophy, and multiple sclerosis (Damsker et al, 2016; Damsker et al, 2013; Dillingham et al, 2015; Heier et al, 2013; Sreetama et al, 2018).”
- “Vamorolone also recently completed Phase 2a trials in DMD patients, where it showed anti-inflammatory efficacy and a dissociation of effects on safety versus efficacy biomarkers in patient serum (Conklin et al, 2018).”

- We cannot perform flow cytometry retroactively on hearts from the MR antagonism trial, however we are looking into developing techniques to do similar assays in diaphragm and heart for future experiments involving *mdx* and receptor knockout mice. We believe these future studies will also provide a cleaner *in vivo* system for studies of inflammation because they will be done without the context of hyperaldosteronism introduced by osmotic pumps in our current study.

Minor comments:

A) The introduction is too long and contains partly repetitions of content.

- We have shortened the Introduction by nearly 20%.

B) It is stated within the result section "Vamorolone shows potent MR antagonist activity in vivo" that "After six weeks, heart function was assayed by echocardiography, blood pressure was measured, and terminal endpoint measures were performed." These data, however, are not presented here.

- As suggested, in the revised manuscript we present this data on blood pressure, echocardiography and terminal endpoint data in Figure 5A (blood pressure), 5C (histology of fibrosis), 5F (heart mass), and 5G-I (echocardiography).

- Blood pressure:

- Fibrosis histology:

- Heart mass:

○ Echocardiography:

C) The steroidal drugs and metabolic derivatives enlisted within the abstract (vamorolone, prednisolone, prednisone, deflazacort) should be clearly referred to its corresponding receptor system they activate (if possible).

• We addressed this by:

- We added some text clarifying the receptors for each ligand in the abstract, though the extent of this was limited by word count limits.
- We also added a Table showing each ligand and its receptors/effects because we felt it would be helpful both to readers and in fully addressing this point. This is found in Table 1. We can remove this Table if the Reviewer or Editor prefer.

▪ Table 1:

		Prednisolone	Deflazacort	Vamorolone	Eplerenone
		Drug effect relative to Prednisone: Blue = beneficial effect, Red = negative side effect			
GR-dependent	Promoter Type:				
	NF-κB	Anti-inflammatory	Anti-inflammatory	Anti-inflammatory	inactive / weak
MR-direct	GRE	Activates	Activates	inactive / weak	inactive / weak
	MRE	Activates	inactive / weak	Antagonist	Antagonist

- D) Could the authors estimate the amount of vamorolone per day that the mice received upon osmotic pump implantation? A clear statement is missing within the results section.
- As suggested, we have clarified this in the Results section and Figure Legends; the mice received vamorolone at 20 mg/kg via daily oral dosing.
 - New text in Results:
 - “Randomized and blinded treatment groups of wild type and *mdx* mice were implanted with subcutaneous osmotic pumps that secreted either vehicle or aldosterone, the physiological MR agonist, at a calculated dose of 0.25 mg/kg/day ($n \geq 8$ per group). The *mdx* mice receiving aldosterone via osmotic pump were also treated with vehicle, vamorolone (20 mg/kg/day), eplerenone (100 mg/kg/day), or spironolactone (20 mg/kg/day), using daily oral administration via ingestion of sugar syrup formulations.”
- E) The antagonistic effect of vamorolone on MR activation in vivo is restricted to maintain the kidney size upon treatment with aldosterone. This aspect of vamorolone is not directly linked to its mode of action at sites of cardiac dystrophy and thus should be transferred to a supplemental material section.
- We agree that this is not directly linked to cardiac dystrophy, however as a traditional MR/aldosterone target tissue it is linked to the mechanism of action of MR ligands. We feel these may actually be viewed as a good reasons to include it, because this MR-mediated phenotype allows us to also assay effects of compounds on an MR activity that is independent of dystrophic pathology. We have decided to keep this in the main text for now because it is directly linked to the drug mechanism of MR antagonism.
 - Additionally, the contrast in effects of prednisolone/aldosterone between kidney and heart is important to address Reviewer 2 Point (A). By showing the effects in both kidney and heart, we can show how heart is specifically sensitized to MR-activation by prednisolone due to its lack of expression for the protective, steroid-metabolizing enzyme HSD11B2.
- F) The figure legends should not contain a description and interpretation of the results itself, e.g. "Vamorolone acted as an MR antagonist, consistent with eplerenone and spironolactone" (Fig. 1) and "Prednisolone caused an increase in D2-mdx heart size" (Fig.5).
- We used this descriptive style because we thought it was consistent with other papers in this journal and the directions in the Life Science Alliance guidelines. We can change this if the Editor prefers.
- G) The authors should employ a more unbiased and global gene expression profiling approach than selective qRT-PCR to characterize changes of gene expression in heart tissues of mdx and vamorolone treated mdx mice.

- In the future, we will look into performing similar profiling experiments towards the development of MR-specific biomarkers. For now, this is beyond the scope of the current manuscript, as the Editor stated in their response that “we don’t expect global gene expression analysis in heart tissues of the mdx mice for acceptance here”.

January 18, 2019

RE: Life Science Alliance Manuscript #LSA-2018-00186-TR

Dr. Christopher R. Heier
Children's National Medical Center
Research Center for Genetic Medicine
111 Michigan Ave NW
Washington, DC 20010

Dear Dr. Heier,

Thank you for submitting your revised manuscript entitled "Vamorolone targets dual nuclear receptors to treat inflammation and dystrophic cardiomyopathy". As you will see, the reviewers appreciate the introduced changes and we would thus be happy to publish your paper in Life Science Alliance pending final revisions necessary to meet our formatting guidelines:

- Fig3 panel E misspelled in the legend as panel D, please correct

A. FINAL FILES:

-- High-resolution figure, supplementary figure and video files uploaded as individual files: See our detailed guidelines for preparing your production-ready images, <http://life-science-alliance.org/authorguide>

B. MANUSCRIPT ORGANIZATION AND FORMATTING:

Full guidelines are available on our Instructions for Authors page, <http://life-science-alliance.org/authorguide>

Sincerely,

Reviewer #1 (Comments to the Authors (Required)):

The authors have addressed my concerns and I congratulate them on an interesting study that is important to the field of muscular dystrophy and beyond

Reviewer #3 (Comments to the Authors (Required)):

The revised manuscript addresses the comments of all of the previous reviews.

January 28, 2019

RE: Life Science Alliance Manuscript #LSA-2018-00186-TRR

Dr. Christopher R. Heier
Children's National Medical Center
Research Center for Genetic Medicine
111 Michigan Ave NW
Washington, DC 20010

Dear Dr. Heier,

Thank you for submitting your Research Article entitled "Vamorolone targets dual nuclear receptors to treat inflammation and dystrophic cardiomyopathy". It is a pleasure to let you know that your manuscript is now accepted for publication in Life Science Alliance. Congratulations on this interesting work.

*****IMPORTANT:** If you will be unreachable at any time, please provide us with the email address of an alternate author. Failure to respond to routine queries may lead to unavoidable delays in publication.*******

DISTRIBUTION OF MATERIALS:

Again, congratulations on a very nice paper. I hope you found the review process to be constructive and are pleased with how the manuscript was handled editorially. We look forward to future exciting submissions from your lab.

Sincerely,
